# Complex-based Ligand-Binding Proteins Redesign by Equivariant Diffusion-based Generative Models

## Abstract

Proteins, serving as the fundamental architects of biological processes, interact with ligands to perform a myriad of functions essential for life. Designing functional ligand-binding proteins is pivotal for advancing drug development and enhancing therapeutic efficacy. In this study, we introduce ProteinReDiff, an efficient computational framework targeting the redesign of ligand-binding proteins. Using equivariant diffusion-based generative models, ProteinReDiff enables the creation of high-affinity ligand-binding proteins without the need for detailed structural information, leveraging instead the potential of initial protein sequences and ligand SMILES strings. Our evaluations across sequence diversity, structural preservation, and ligand binding affinity underscore ProteinReDiff's potential to advance computational drug discovery and protein engineering. We will release our data and source code upon acceptance.

## 1 Introduction

Proteins, often referred to as the molecular architects of life, play a critical role in virtually all biological processes. A significant portion of these functions involves interactions between proteins and ligands, underpinning the complex network of cellular activities. These interactions are not only pivotal for basic physiological processes, such as signal transduction and enzymatic catalysis, but also have broad implications in the development of therapeutic agents, diagnostic tools, and various biotechnological applications (Du et al., 2016; Wanat, 2020; Skolnick & Zhou, 2022). Despite the paramount importance of protein-ligand interactions, the majority of existing studies have primarily focused on protein-centric designs to optimize specific protein properties, such as stability, expression levels, and specificity (Listov et al., 2024; Lisanza et al., 2023; Dauparas et al., 2022; Yang et al., 2023a; Iqbal et al., 2022). This prevalent approach, despite leading to numerous advancements, does not fully exploit the synergistic potential of optimizing both proteins and ligands for redesigning ligand-binding proteins. By embracing an integrated design approach, it becomes feasible to refine control over binding affinity and specificity, leading to applications such as tailored therapeutics with reduced side effects, highly sensitive diagnostic tools, efficient biocatalysis, targeted drug delivery systems, and sustainable bioremediation solutions (Yang & Lai, 2017; Ebrahimi & Samanta, 2023; Ruscito & DeRosa, 2016), thus illustrating the transformative impact of redesigning ligand-binding proteins across various fields.

Traditional methods for designing ligand-binding proteins have relied heavily on experimental techniques, characterized by systematic but often inefficient trial-and-error processes (Creutznacher et al., 2022; Munk et al., 2016; Tavares & van der Meer, 2021). These methods, while foundational, are time-consuming, resource-intensive, and sometimes fall short in precision and efficiency. The emergence of computational design has marked a transformative shift, offering new pathways to accelerate the design process and gain deeper insights into the molecular basis of protein-ligand interactions. However, even with the advancements in computational approaches, significant challenges remain. Many existing models demand extensive structural information, such as protein crystal structures and specific binding pocket data, limiting their applicability, especially in urgent scenarios like the emergence of novel diseases (Polizzi & DeGrado, 2020; Stärk et al., 2023; Dauparas et al., 2023). For instance, during the outbreak of a new disease like COVID-19, the spike proteins of the virus may not have well-characterized binding sites, delaying the development

of effective drugs (Lv et al., 2020; Schaub et al., 2021). Furthermore, the complexity of binding mechanisms, including allosteric effects and cryptic pockets, adds another layer of difficulty (Agajanian et al., 2023; Oleinikovas et al., 2016). Specifically, many proteins do not exhibit clear binding pockets until ligands are in close vicinity, necessitating extensive simulations to reveal potential binding interfaces (Meller et al., 2023; Oleinikovas et al., 2016). While molecular dynamics simulations offer detailed atomistic insights into binding mechanisms, they often prove inadequate for designing high-throughput sequences due to high computational cost (Barros et al., 2019; Yang & Lai, 2017). This complexity underscores the need for a drug design methodology that is agnostic to predefined binding pockets.

Our study addresses those identified challenges by introducing ProteinReDiff, a computational framework developed to enhance the process of redesigning ligand-binding proteins. Originating from the foundational concepts of the Equivariant Diffusion-Based Generative Model for Protein-Ligand Complexes (DPL) (Nakata et al., 2023), ProteinReDiff incorporates key improvements inspired by the representation learning modules from the AlphaFold2 (AF2) architecture (Jumper et al., 2021). Specifically, we integrate the Outer Product Update (adapted from outer product mean of AF2), Single Representation Attention (adapted from MSA row attention module), and Triangle Multiplicative Update modules into our Residual Feature Update procedure. These modules collectively enhance the framework's ability to capture intricate protein-ligand interactions, improve the fidelity of binding affinity predictions, and enable more precise redesigns of ligand-binding proteins.

The framework integrates the generation of diverse protein sequences with blind docking capabilities. Starting with a selected protein-ligand pair, our approach stochastically masks amino acids and equivariantly denoises the diffusion model to capture the joint distribution of ligand and protein complex conformations. Another key feature of our method is blind docking, which predicts how the redesigned protein interacts with its ligand without the need for predefined binding site information, while relying solely on initial protein sequences and ligand SMILES strings (Weininger, 1988). This streamlined approach significantly reduces reliance on detailed structural data, thus expanding the scope for sequence-based exploration of protein-ligand interactions.

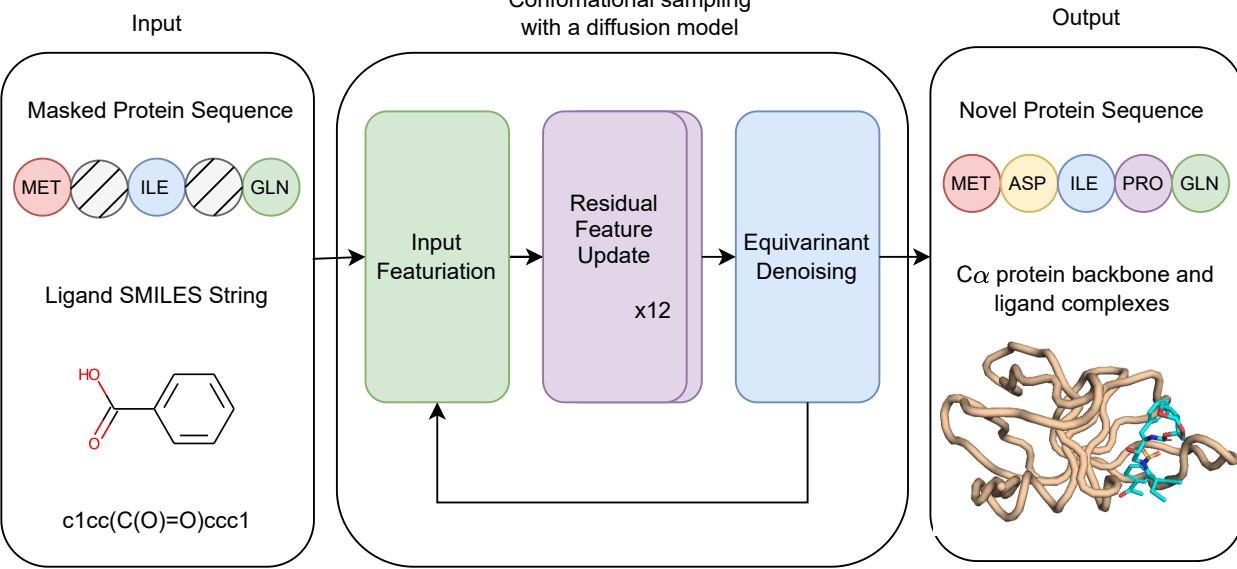

Figure 1: Overview of the proposed framework. The process begins with utilizing a protein amino acid sequence and a ligand SMILES string as inputs. The conformational sampling process includes iteratively applying input featurization, updating residual features, and denoising equivariantly, ultimately yielding novel protein sequences alongside their corresponding $C\alpha$ protein backbone and ligand complexes.

In summary, the contributions of our paper are outlined as follows:

- We introduce ProteinReDiff, an efficient computational framework for ligand-binding protein redesign, rooted in equivariant diffusion-based generative models. Our innovation lies in integrating AF2's representational learning modules to enhance the framework's ability to capture intricate protein-ligand interactions.

- Our framework enables the design of high-affinity ligand-binding proteins without reliance on detailed structural information, relying solely on initial protein sequences and ligand SMILES strings.

- We comprehensively evaluate our model's outcomes across multiple design aspects, including sequence diversity, structure preservation, and ligand binding affinity, ensuring a holistic assessment of its effectiveness and applicability in various contexts.

## 2 Related Work

### 2.1 Traditional Approaches in Protein Design

Protein design has historically hinged on computational and experimental strategies that paved the way for modern advancements in the field. These foundational methodologies emphasized the intricate balance between understanding protein structure and engineering novel functionalities, albeit with inherent limitations in scalability and precision. Key traditional approaches include:

- **Rational Design** Korendovych (2018); Song et al. (2023); Alley et al. (2019) focused on introducing specific mutations into proteins based on known structural and functional insights. This method required an in-depth understanding of the target protein structures and how changes might impact its function.

- **Directed Evolution** Arnold & Volkov (1999); Wang & Zhao (2016); Guntas et al. (2005); Waltenspühl et al. (2021) mimicked natural selection in the laboratory, evolving proteins towards desired traits through iterative rounds of mutation and selection. Despite its effectiveness in discovering functional proteins, the process was often labor-intensive and time-consuming.

These traditional methods have been instrumental in advancing our understanding and capability in protein design. However, their limitations in terms of efficiency, specificity, and the broad applicability of findings highlighted the need for more versatile and scalable approaches. As the field progressed, the integration of computational power and biological understanding opened new avenues for innovation in protein design, leading to the exploration and adoption of more advanced methodologies.

### 2.2 Deep Generative Models in Protein Design

Since their inception, deep generative models have significantly advanced fields like computer vision (CV) (Raut & Singh, 2024) and natural language processing (NLP) (Iqbal & Qureshi, 2022), sparking interest in their application to protein design. This enthusiasm has led to numerous studies that harness these models for innovating within the protein design area. Among these, certain types of deep generative models have distinguished themselves through their effectiveness and the promising results they have achieved, including:

- **Variational Autoencoders (VAEs)** are harnessed for their ability to learn rich representations of protein sequences, enabling the generation of novel sequences through manipulation in the latent space (Lyu et al., 2023; Greener et al., 2018; Brookes et al., 2019).

- **Autoregressive models** predict the probability of each amino acid in a sequential manner, facilitating the generation of coherent and functionally plausible protein sequences (Trinquier et al., 2021; Fannjiang et al., 2022).

- **Generative Adversarial Networks (GANs)** employ two networks that work in tandem to produce protein sequences indistinguishable from real ones, enhancing the realism and diversity of generated designs (Kucera et al., 2022; Anand & Huang, 2018).

- **Diffusion models** represent a step forward by gradually transforming noise into structured data, simulating the complex process of folding sequences into functional proteins (Gruver et al., 2023; Watson et al., 2023; Wu et al., 2022a; Fu et al., 2023).

However, the majority of these studies have focused on protein-centric designs, with a noticeable gap in research that integrates both proteins and ligands for the purpose of redesigning ligand-binding proteins. Such integration is crucial for a holistic understanding of the intricate dynamics between protein structures and their ligands, a domain that remains underexplored.

### 2.3 Current Approaches in Ligand-Binding Protein Redesign

**Heavy Reliance on Detailed Structural Information**  Contemporary computational methodologies for designing proteins that target specific surfaces predominantly rely on structural insights from native complexes, underscoring the critical role of fine-tuning side-chain interactions and optimizing backbone configurations for optimal binding affinity (Polizzi & DeGrado, 2020; Stärk et al., 2023; Dauparas et al., 2023; Zheng et al., 2023; Watson et al., 2023; Yang et al., 2022). These strategies often initiate with the generation of protein backbones, employing inverse folding techniques to identify sequences capable of folding into these pre-designed structures (Yang et al., 2023a; Hsu et al., 2022; Dauparas et al., 2022; Yang et al., 2022). This approach signifies a paradigm shift by prioritizing structural prediction ahead of sequence identification, aiming to produce proteins that not only fit the desired conformations for potential ligand interactions but also navigate around the challenge of undefined binding sites. Despite the advantages, including the potential of computational docking to create binders via manipulation of antibody scaffolds and varied loop geometries (Lyu et al., 2023; Bennett et al., 2024; Chungyoun & Gray, 2023), a notable challenge persists in validating these binding modes with high-resolution structural evidence. Additionally, the traditional focus on a limited array of hotspot residues for guiding protein scaffold placement often restricts the exploration of possible interaction modes, particularly in cases where target proteins lack clear pockets or clefts for ligand accommodation (Meller et al., 2023; Gagliardi & Rocchia, 2023).

**Limited Training Data and Lack of Diversity**  Existing approaches often rely on a limited set of training data, which can restrict the diversity and generalizability of the resulting models. For instance, datasets like PDBBind provide detailed ligand information, but their scope is limited (Wang et al., 2004). This limitation is further compounded when protein datasets lack corresponding ligand data, reducing the effectiveness of the training process. Traditional methodologies also tend to focus on a narrow range of protein-ligand interactions, potentially overlooking the broader spectrum of possible interactions.

**Single-Domain Denoising Focus**  Previous methodologies typically concentrate on denoising either in sequence space or structural space, but not both. Approaches like ProteinMPNN (Dauparas et al., 2022), LigandMPNN (Dauparas et al., 2023), and MIF (Yang et al., 2022) primarily operate in sequence space, while others like DPL function in structural space (Nakata et al., 2023). This single-domain focus can limit the ability to capture the full complexity of protein-ligand interactions, which inherently involve both sequence and structural dimensions. Consequently, these methodologies may fall short of accurately predicting the functional capabilities of redesigned proteins.

**Challenges in Generating Diverse Sequences with Structural Integrity**  While some approaches prioritize sequence similarity to generate functional proteins, they often do so at the expense of structural integrity. For example, ProteinMPNN and CARP focus heavily on sequence similarity, which can result in a lack of diversity and flexibility in the generated sequences (Dauparas et al., 2022; Yang et al., 2023a). This limitation can hinder the ability to explore a wider range of functional conformations, reducing the effectiveness of the protein design process.

**Distinct Improvements of Our Approach**  We address the weaknesses of available methodologies by integrating diverse datasets, employing a dual-domain denoising strategy, and ensuring the generation of diverse sequences while maintaining structural integrity. Our approach utilizes only protein sequences and ligand SMILES strings, eliminating the need for detailed structural information. By combining PDBBind (Wang

et al., 2004) and CATH (Sillitoe et al., 2018) datasets, we effectively double our training data, enhancing protein representations. Our equivariant and KL-divergence loss functions enable denoising across both sequence and structural dimensions, capturing the full complexity of protein-ligand interactions. This approach maintains structural fidelity and promotes sequence diversity, overcoming the limitations of methodologies prioritizing sequence similarity at the expense of diversity.

## 3 Background

### 3.1 Protein Language Models (PLMs)

Protein Language Models (PLMs) harness the power of natural language processing (NLP) to unravel the intricate latency embedded within protein sequences. By analogizing amino acid sequences to human language sentences, PLMs unlock profound insights into protein functions, interactions, and evolutionary trajectories (Yang et al., 2023b). These models leverage advanced text processing techniques to predict structural, functional, and interactional properties of proteins based solely on their amino acid sequences (Brandes et al., 2022; Elnaggar et al., 2022; Rives et al., 2021; Lin et al., 2023a). Their adoption in protein design has catalyzed significant progress, with studies leveraging PLMs to translate protein sequence data (Madani et al., 2023; Ruffolo & Madani, 2024; Min et al., 2024; Zheng et al., 2023) into actionable insights, thus guiding the precise engineering of proteins with targeted functional attributes.

Mathematically, a PLM can be represented as a function $F$ that maps a sequence of amino acids $S = [s_1, s_2, \ldots, s_n]$, where $s_i$ denotes the $i$-th amino acid in the sequence, to a high-dimensional feature space that encapsulates the protein's structural and functional properties:

$$X = F(S), \quad X \in R^d, \tag{1}$$

where $X$ represents the continuous representation or embedding derived from the sequence $S$ and $d$ represents the dimensionality of the embedding space, determined by the PLM's architecture. This embedding captures the complex dependencies and patterns underlying the protein's structural information and biological functionality. Through training on known sequences and structures, PLMs discern the "grammar" governing protein folding and function, facilitating accurate predictions.

We employ the ESM-2 model (Lin et al., 2023a), a state-of-the-art protein language model with 650 million parameters, pre-trained on nearly 65 million unique protein sequences from the UniRef (Suzek et al., 2014) database, to feature initial masked protein sequences. ESM-2 enriches the latent representation of protein sequences, bypassing the need for conventional multiple sequence alignment (MSA) methods. By incorporating structural and evolutionary information from input sequences, ESM-2 enables us to unravel interaction patterns across protein families for effective ligand targeting. This understanding is crucial for designing and optimizing ligand-binding proteins.

### 3.2 Equivariant Diffusion-based Generative Models

We utilize a generative model driven by equivariant diffusion principles, drawing from the foundations laid by Variational Diffusion Models (Kingma et al., 2023) and E(3) Equivariant Diffusion Models (Hoogeboom et al., 2022).

#### 3.2.1 The Diffusion Procedure

First, we employ a diffusion procedure that is equivariant with respect to the coordinates of atoms $x$, alongside a series of progressively more perturbed versions of $x$, known as latent variables $z_t$, with $t$ varying from 0 to 1. To maintain translational invariance within the distributions, we opt for distributions on a linear subspace that anchors the centroid of the molecular structure at the origin, and designate $N_x$ as a Gaussian distribution within this specific subspace. The conditional distribution of the latent variable $z_t$ given $x$, for any given $t$ in the interval [0, 1], is defined as

$$q(z_t|x) = N_x(\alpha_t x, \sigma_t^2 I), \tag{2}$$

where $\alpha_t$ and $\sigma_t^2$ represent strictly positive scalar functions of $t$, dictating the extent of signal preservation versus noise introduction, respectively. We implement a variance-conserving mechanism where $\alpha_t = 1 - \sigma_t^2$ and posit that $\alpha_t$ smoothly and monotonically decreases with $t$, ensuring $\alpha_0 \approx 1$ and $\alpha_1 \approx 0$. Given the Markov property of this diffusion process, it can be described via transition distributions as

$$q(z_t|z_s) = N_x(\alpha_{t|s}z_s, \sigma_{t|s}^2 I), \tag{3}$$

for any $t > s$, where $\alpha_{t|s} = \alpha_t/\alpha_s$ and $\sigma_{t|s}^2 = \sigma_t^2 - \alpha_t^2\sigma_s^2$. The Gaussian posterior of these transitions, conditional on $x$, can be derived using Bayes' theorem:

$$q(z_s|z_t, x) = N_x(\mu_{t \to s}(z_t, x), \sigma_{t \to s}^2 I), \tag{4}$$

with

$$\mu_{t \to s} = \frac{\alpha_s\sigma_{t|s}^2}{\alpha_{t|s}\sigma_s^2}z_t + \frac{\sigma_s^2\sigma_t^2}{\sigma_{t|s}^2}x, \quad \sigma_{t \to s}^2 = \frac{\sigma_t^2\sigma_s^2}{\sigma_{t|s}^2}. \tag{5}$$

### 3.2.2 The Generative Denoising Process

The construction of the generative model inversely mirrors the diffusion process, generating a reverse temporal sequence of latent variables $z_t$ from $t = 1$ back to $t = 0$. By dividing time into $T$ equal intervals, the generative framework can be described as:

$$p_\theta(x) = \int_z p(z_1)p(x|z_0)\prod_{i=1}^{T} p_\theta(z_{t_i}|z_{t_{i-1}}), \tag{6}$$

with $s(i) = (i-1)/T$ and $t(i) = i/T$. Leveraging the variance-conserving nature and the premise that $\alpha_1 \approx 0$, we posit $q(z_1) = N_x(0, I)$, hence treating the initial distribution of $z_1$ as a standard Gaussian:

$$p(z_1) = N_x(0, I). \tag{7}$$

Furthermore, under the variance-preserving framework and assuming $\alpha_0 \approx 1$, the distribution $q(z_0 \mid x)$ is modeled as highly peaked (Song et al., 2020; Kingma et al., 2023). This allows us to approximate $p_{\text{data}}(x)$ as nearly constant within this narrow peak region. This yields:

$$q(x|z_0) = \frac{q(z_0|x)p_{\text{data}}(x)}{\int_{\tilde{x}} q(z_0|\tilde{x})p_{\text{data}}(\tilde{x})} \approx \frac{q(z_0|x)}{\int_{\tilde{x}} q(z_0|\tilde{x})} = \mathcal{N}_x(x|z_0/\alpha_0, \sigma_0^2/\alpha_0^2 I). \tag{8}$$

Accordingly, we approximate $q(x|z_0)$ through:

$$p(x|z_0) = \mathcal{N}_x(x|z_0/\alpha_0, \sigma_0^2/\alpha_0^2 I). \tag{9}$$

The generative model's conditional distributions are then formulated as:

$$p_\theta(z_s|z_t) = q(z_s|z_t, x = \hat{x}_\theta(z_t; t)), \tag{10}$$

which mirrors $q(z_s|Fsz_t, x)$ but substitutes the actual coordinates $x$ with the estimates from a temporal denoising model $\hat{x}_\theta(z_t; t)$, which employs a neural network parameterized by $\theta$ to predict $x$ from its noisier version $z_t$. This denoising model's framework, predicated on noise prediction $\hat{\epsilon}_\theta(z_t; t)$, is articulated as:

$$\hat{x}_\theta(z_t; t) = \frac{(z_t - \sigma_t\hat{\epsilon}_\theta(z_t; t))}{\alpha_t}. \tag{11}$$

Consequently, the transition mean $\mu_{t \to s}(z_t, \hat{x}_\theta(z_t; t))$ is determined by:

$$\mu_{t \to s}(z_t, \hat{x}_\theta(z_t; t)) = \frac{\alpha_s\sigma_{t|s}^2}{\alpha_{t|s}\sigma_s^2}z_t + \frac{\alpha_s\sigma_t^2}{\sigma_{t|s}^2}x = \frac{1}{\alpha_{t|s}}z_t - \frac{\sigma_{t|s}^2}{\alpha_{t|s}\sigma_t}\hat{\epsilon}_\theta(z_t; t). \tag{12}$$

# 4 Method

In this section, we detail the methodology employed in our noise prediction model, which is depicted in Figure 1 and consists of three main procedures: (1) input featurization, (2) residual feature update, and (3) equivariant denoising. Through these steps, we transform raw protein and ligand data into structured representations, iteratively refine their features, and leverage denoising techniques inherent in the diffusion model to improve sampling quality.

## 4.1 Input Featurization

We develop both single and pair representations from protein sequences and ligand SMILES string (Figure 2). For proteins, we initially applied stochastic masking to segments of the amino acid sequences. The protein representation is attained through the normalization and linear mapping of the output from the final layer of the ESM-2 model, which is subsequently combined with the amino acid and masked token embeddings. Additionally, for pair representations of proteins, we leveraged pairwise relative positional encoding techniques, drawing from established methodologies (Jumper et al., 2021). For ligand representations, we employed a comprehensive feature embedding approach, capturing atomic and bond properties such as atomic number, chirality, connectivity, formal charge, hydrogen attachment count, radical electron count, hybridization status, aromaticity, and ring presence for atoms; and bond type, stereochemistry, and conjugation status for bonds. These representations are subsequently merged, incorporating radial basis function embeddings of atomic distances and sinusoidal embeddings of diffusion times. Together, these steps culminate in the formation of preliminary complex representations, laying the foundation for our computational analyses.

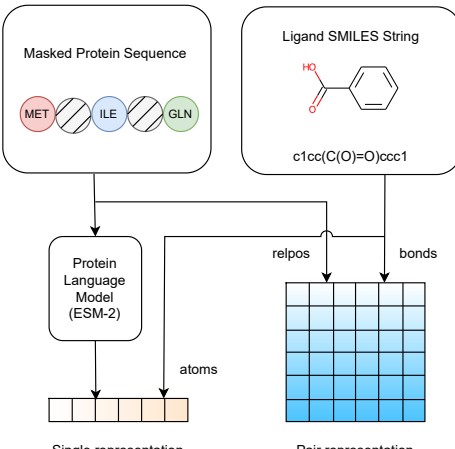

Figure 2: Overview of the input featurization procedure of the model.

## 4.2 Residual Feature Update Procedure

Our approach deviates significantly from the residual feature update procedure employed in the original DPL model (Nakata et al., 2023). While the DPL model relied on Alphafold2's Triangular Multiplicative Update for updating single and pair representations, where these representations mutually influence each other, our objective is to optimize this procedure for greater efficiency. Specifically, we incorporate enhancements such as the Outer Product Update and Single Representation Attention to formulate sequence representational hypotheses of protein structures and to model suitable motifs for binding target ligands specifically. These modules, integral to Evoformer, the sequence-based module of AF2, play a crucial role in extracting essential connections among internal motifs that serve structural functions (i.e., ligand binding) when structural information is not explicitly provided during training. Importantly, we adapt and tailor these modules to fit within our model architecture, ensuring their effectiveness in capturing the intricate interplay between proteins and ligands.

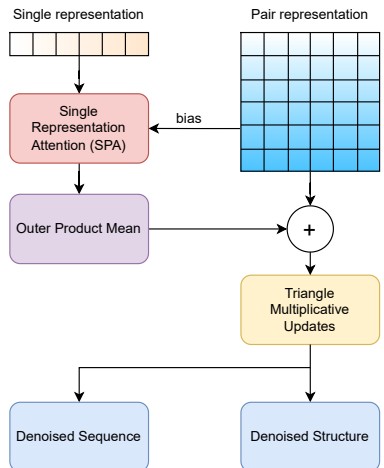

Figure 3: Overview of the residual feature update procedure of the model.

### 4.2.1 Single Representation Attention Module

The Single Representation Attention (SRA) module, derived from the Alphafold2 model's MSA row attention with pair bias, accounts for long-range interactions among residues and ligand atoms within a single protein-ligand embedding vector. In essence, the attention mechanism assigns importance to those involved in complex-based folding to denoise the equivariant loss (Section 4.3) in a self-supervised manner. While the original Alphafold2 MSA row attention mechanism processes input for a single sequence, the SRA module is designed to incorporate representations from multiple protein-ligand complexes concurrently. Specifically, the pair bias component of the SRA attention module captures dependencies between proteins and ligands, which was shown to fit the attention score better than the regular self-attention model without bias terms (Xu et al., 2023). By considering both the single representation vector (which encodes the protein/ligand sequential representation) and the pairwise representation vector (which encodes protein-protein and protein-ligand interactions), this cross-attention mechanism exchanges information between pairwise and single representation to effectively preserves internal motifs, as evidenced by contact overlap metrics (Rao et al., 2021; Yang et al., 2023b). As transformer architecture is widely used for predicting protein functions (Buton et al., 2023), we observed similar efficacy to our binding affinity prediction in section Results 5.2.5 and Appendix B-C.2. For a detailed description of the computational steps implemented in this module, refer to Algorithm 1.

---

**Algorithm 1** Single Representation Attention pseudocode

**Input:** Single representation vector $m_{si}$, pair representation vector $z_{sij}$ of the $i$-th sequence in the set of sequences $s$, C = 65, $N_{head} = 4$.

**Output:** Updated single representation vector $\tilde{m}_{si}$ with the dimension of $C_m$.

1: $m_{si} \leftarrow \text{LayerNorm}(m_{si})$
2: $q_{si}^h, k_{si}^h, v_{si}^h \leftarrow \text{LinearNoBias}(m_{si})$   $q_{si}^h, k_{si}^h, v_{si}^h \in \mathbb{R}^C, h \in \{1, \ldots, N_{\text{head}}\}$
3: $b_{sij}^h \leftarrow \text{LinearNoBias}(\text{LayerNorm}(z_{sij}))$
4: $g_{si}^h \leftarrow \text{sigmoid}(\text{Linear}(m_{si}))$   $g_{si}^h \in \mathbb{R}^C$
5: $a_{sij}^h \leftarrow \text{softmax}_j \left( \frac{1}{\sqrt{C}} q_{si}^h {k_{sj}^h}^T + b_{sij}^h \right)$
6: $o_{si}^h \leftarrow g_{si}^h \odot \sum_j a_{sij}^h v_{sj}^h$
7: $\tilde{m}_{si} \leftarrow \text{Linear}(\text{concat}_h(o_{si}^h))$   $\tilde{m}_{si} \in \mathbb{R}^{C_m}$
8: return$\{\tilde{m}_{si}\}$

---

### 4.2.2 Outer Product Update

Since the SRA encodings have a shape $(s, r, c_m)$ and the pair representation has a shape $(s, r, r, c_z)$, the outer product (OPU) layer merges insights by reshaping SRA encodings into pair representations. This module leverages evolutionary cues from ESM to generate plausible structural hypotheses for pair representations (Ju et al., 2021). It first calculates the outer product of the SRA embeddings of protein-ligand pairs, then aggregates the outer products to yield a measure of co-evolution between every residue pair (Yang et al., 2023b). Analogous to Tensor Product Representations (TPR) in NLP, the outer product is akin to the filler-and-role binding relationship, where each entity (i.e. amino acid residue) on a sequence is attached to a rich functional embedding based on its relationship to one another (Huang et al., 2018; Smolensky, 1990; Huang et al., 2019).

This process integrates correlated information of residues $i$ and $j$ of a sequence $s$, resulting in the intermediate Kronecker product tensors (.i.e. role embeddings in NLP) (Xu et al., 2023; Schlag & Schmidhuber, 2018; Chen et al., 2021). Subsequently, an affine transformation projects those representations to hypotheses concerning the relative positions of residues $i$ and $j$ under biophysical constraints. Our implementation adapts the outer product without computing the mean to maintain the pair representations of multiple protein-ligand complexes. For a detailed description of the computational steps implemented in this module, refer to Algorithm 2.

---

**Algorithm 2** Outer product update pseudocode

**Input:** Single representation vector $m_{si}$ of the $i$-th sequence in the set of sequences $s$, $C = 32$.
**Output:** Pair representation vector $z_{sij}$ with the dimension of $s \times C_z$.

1: $m_{si} \leftarrow \text{LayerNorm}(m_{si})$
2: $a_{si}, b_{si} \leftarrow \text{Linear}(m_{si}) \quad a_{si}, b_{si} \in \mathbb{R}^C$
3: $o_{sij} \leftarrow \text{flatten}(a_{si} \otimes b_{si}) \quad o_{sij} \in \mathbb{R}^{C \times C}$
4: $z_{sij} \leftarrow \text{Linear}(o_{sij}) \quad z_{sij} \in \mathbb{R}^{s \times C_z}$
5: $\text{return}\{z_{sij}\}$

---

### 4.2.3 Triangle Multiplicative Updates

After refining the pair representation, our model interprets the primary protein-ligand structure using principles from graph theory, treating each residue as a distinct entity interconnected through the pairwise matrix. These connections are then refined through triangular multiplicative updates to account for physical and geometric constraints, such as triangular inequality. While the SRA weights the importance of residues, the triangular multiplicative update acts as another stack of transformer-based layers where any two edges affect the third one to enforce triangle equivariance (Lin & AlQuraishi, 2023; Yang et al., 2023b). The starting and ending nodes propagate information in and out of neighbors in similar fashion as the message-passing framework (Xu et al., 2023). These mechanisms enable the model to generate more accurate representations of protein-ligand complexes, leading to improved predictive performance in predicting binding affinities and structural characteristics.

### 4.3 Equivariant Denoising

During the equivariant denoising process, the final pair representation undergoes symmetrization and is then transformed using a multi-layer perceptron (MLP) into a weight matrix $W$. This matrix is utilized to compute the weighted sum of all relative differences in 3D space for each atom, as shown in the equation (Nakata et al., 2023):

$$\hat{\epsilon}_i(z) = \sum_j W_{ij}(z) \cdot \frac{(z_i - z_j)}{\|z_i - z_j\|}. \tag{13}$$

Afterward, the centroid is subtracted from this computation, resulting in the output of our noise prediction model $\hat{\epsilon}$. Additionally, it's important to note that the described model maintains SE(3)-equivariance, meaning

that:

$$\hat{\epsilon}_i(\mathbf{R}z + \mathbf{t}) = \sum_j \frac{W_{ij}(\mathbf{R}z + \mathbf{t})}{\|(\mathbf{R}z_i + \mathbf{t}) - (\mathbf{R}z_j + \mathbf{t})\|} \cdot ((\mathbf{R}z_i + \mathbf{t}) - (\mathbf{R}z_j + \mathbf{t})) \tag{14}$$

$$= \mathbf{R} \sum_j \frac{W_{ij}(\mathbf{R}z + \mathbf{t})}{\|z_i - z_j\|} \cdot (z_i - z_j) \tag{15}$$

$$= \mathbf{R} \sum_j \frac{W_{ij}(z)}{\|z_i - z_j\|} \cdot (z_i - z_j) \tag{16}$$

$$= \mathbf{R}\hat{\epsilon}_i(z) \tag{17}$$

for any rotation $\mathbf{R}$ and translation $\mathbf{t}$. This property is derived from the fact that the final representation, and hence the weight matrix $W$, depends solely on atom distances that are invariant to rotation and translation.

## 5 Experiments

### 5.1 Training Process

#### 5.1.1 Materials

Our training strategy leverages a meticulously curated dataset encompassing a broad range of protein structures, including both ligand-bound (holo) and ligand-free (apo) forms, sourced from two key repositories: PDBBind v2020 (Wang et al., 2004) and CATH 4.2 (Sillitoe et al., 2018). PDBBind v2020 offers a diverse collection of protein-ligand complexes, while CATH 4.2 provides a substantial repository of protein structures. Each dataset was selected for its unique contributions to our understanding of protein-ligand interactions and structural diversity. This strategic selection of datasets ensures our model is exposed to a wide and varied spectrum of protein-ligand interactions and structural configurations, enabling comprehensive evaluation against diverse inverse folding benchmarks. By training on both holo and apo structures, our approach imbues the model with a robust understanding of protein-ligand dynamics, equipping it to navigate the complexities of unseen protein-ligand interaction scenarios effectively.

To ensure robust model training and evaluation, we employ careful data partitioning techniques. Using MMseqs2 (Steinegger & Söding, 2017), we clustered and partitioned the protein sets for training, validation, and testing, maintaining sequence similarities between 40% and 50% to ensure unbiased training and predictions, following protocols from other protein models (Yang et al., 2022; Jumper et al., 2021). For ligands, we cluster based on the Tanimoto similarity of Morgan fingerprints (Morgan, 1965) on ligand structures. Incorporating CATH 4.2 data into PDBBind not only preserves the objectivity of the train/test/validation partitions but also substantially decreases the similarities within ligand sets, as shown in Table 1.

Table 1: Similarity between Train/Validation/Test Sets of Proteins and Ligands. The values represent similarity percentages for the original PDBBind dataset versus combined PDBBind with CATH datasets in parentheses.

| Protein | Validation | Test | Ligand | Validation | Test |
|---|---|---|---|---|---|
| Train | 36.0% (36.2%) | 38.0% (42.2%) | Train | 72.2% (36.1%) | 9.41% (3.11%) |
| Validation | - | 39.08% (43.5%) | Validation | - | 9.37% (3.17%) |

Table 2 provides an overview of the partitioning details, facilitating a clear understanding of the distribution of samples across different subsets of the dataset.

- **PDBBind v2020**: For consistency and comparability with previous studies, we first adhered to the test/training/validation split settings outlined in established literature (Ingraham et al., 2019), specifically following the configurations defined in the respective sources for the PDBBind v2020

Table 2: Data Partitioning Overview (Unit: number of samples)

| Dataset | Train | Validation | Test |
|---------|-------|------------|------|
| PDBBind v2020 | 9430 | 552 | 207 |
| CATH 4.2 | 15261 | 939 | - |

datasets (Koh et al., 2023). Then, we filtered out those highly similar sequences (above 95%) to keep the average similarities between 40%-50%.

- **CATH 4.2**: In our approach, we deliberately focused on proteins with fewer than 400 amino acids and less similar (below 90%) sequences from the CATH 4.2 database. This selective criterion was chosen to prioritize smaller proteins, which often represent more druggable targets of interest in drug discovery and development endeavors. During both the training and validation phases, SMILES strings of CATH 4.2 proteins were represented as asterisks (masked tokens) to denote unspecified ligands. Notably, CATH 4.2 was excluded from the test set due to the absence of corresponding ligands required for evaluating protein-ligand interactions.

### 5.1.2 Loss Functions

Previous models typically denoise in only one domain, such as ProteinMPNN (Dauparas et al., 2022), LigandMPNN (Dauparas et al., 2023), and MIF (Yang et al., 2022) in sequence space, and DPL (Nakata et al., 2023) in structural space. This limitation restricts their ability to fully capture the intricate interactions between proteins and ligands. To address this, we have introduced significant modifications to the loss function to better suit the task of ligand-binding protein redesign. By tailoring the loss function to integrate both sequence and structural spaces, our approach effectively addresses the unique challenges of protein-ligand interactions. Specifically, the optimization of our model for ligand-binding protein redesign is governed by a composite loss function $L$, formulated as follows:

$$L = L_{\text{WS}} + L_{\text{KL}} + L_{\text{CE}}, \tag{18}$$

**Weighted Sum of Relative Differences ($L_{\text{WS}}$)**   This component ensures the model's sensitivity to the directional influence between atoms, supporting the accurate prediction of the denoised structure while maintaining physical symmetries. It is crucial for the equivariant denoising step, enabling accurate noise prediction for atoms in the protein-ligand complex. The loss is defined as:

$$L_{\text{WS}} = \sum_{t=1}^{T} \|\epsilon - \hat{\epsilon}_\theta(z; t)\|, \tag{19}$$

where $T$ is the total number of time steps in the diffusion process, $\epsilon$ is the Gaussian noise vector $\mathcal{N}(\mathbf{0}, \mathbf{I})$, and $\hat{\epsilon}_\theta(z; t)$ is the loss prediction at time step $t$ parameterized by a weight MLP in Section 4.3.

**Kullback-Leibler Divergence ($L_{\text{KL}}$)**   (Joyce, 2011) This component quantifies the divergence between the model's predictions and actual sequence data at timestep $t - 1$, playing a pivotal role in the denoising process. Defined as $KL(x_{\text{pred\_t-1}}, seq_{\text{t-1}})$, it contrasts the predicted distribution, $x_{\text{pred\_t-1}}$, against the true sequence distribution, $seq_{\text{t-1}}$, leveraging the diffusion process's $\gamma$ parameter for temporal adjustment. This loss is also applied in the Protein Generator (Lisanza et al., 2023) model to ensure the model's predictions progressively align with actual data distributions, enhancing the accuracy of sequence and structure generation by minimizing the expected divergence.

**Cross-entropy Loss ($L_{\text{CE}}$)**   This loss function is crucial for the accurate prediction of protein sequences, aligning them with the ground truth through effective classification. It denoises each amino acid from masked latent embedding to a specific class, leveraging categorical cross-entropy to rigorously penalize discrepancies between the model's predicted probability distributions and the actual distributions for each amino acid type.

### 5.1.3 Training Performance

Throughout the training phase, we meticulously observed the model's performance, paying close attention to the dynamics between training and validation losses, as demonstrated in Figure 4. While the training loss consistently diminished, indicating effective learning, the validation loss exhibited more variability. Despite these fluctuations, the validation loss showed an overall downward trend, suggesting that the model is improving its generalization capabilities over time. The general alignment between the downward trends of training and validation losses indicates that the model is learning effectively without significant overfitting.

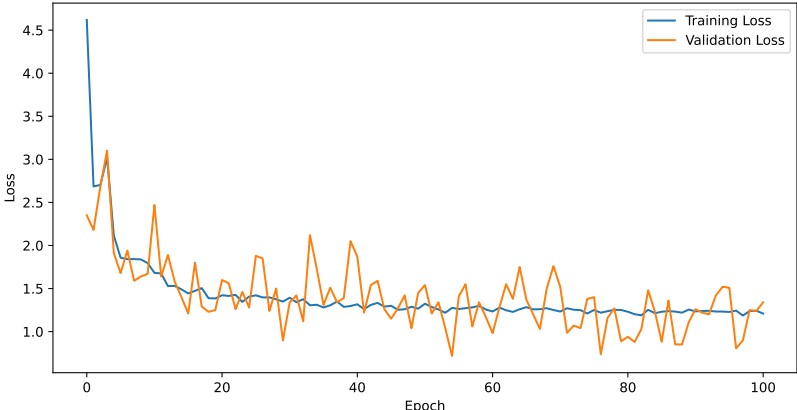

Figure 4: Training history chart of ProteinReDiff, showcasing the evolution of training and validation losses over epochs.

## 5.2 Evaluation Process

### 5.2.1 Ligand Binding Affinity (LBA)

Ligand binding affinity is a fundamental measure that quantifies the strength of the interaction between a protein and a ligand. This metric is crucial as it directly influences the effectiveness and specificity of potential therapeutic agents; higher affinity often translates to increased drug efficacy and lower chances of side effects (Sawada et al., 2024). Within this context, ProteinReDiff is evaluated on its ability to generate protein sequences for significantly improved binding affinity with specific ligands. We utilize a docking score-based approach for this assessment, where the docking score serves as a quantitative indicator of affinity. Expressed in kcal/mol, these scores inversely relate to binding strength — lower scores denote stronger, more desirable binding interactions.

### 5.2.2 Sequence Diversity

Sequence diversity is crucial for exploring protein's functional space (Ziegler et al., 2023). It reflects the capacity of our model, ProteinReDiff, to traverse the vast landscape of protein sequences and generate a wide array of variations. To quantitatively assess this diversity, we utilize the average edit distance (Levenshtein distance) (Miller et al., 2009) between all pairs of sequences generated by the model. This metric offers a nuanced measure of variability, surpassing traditional metrics that may overlook subtle yet significant differences. The diversity score is calculated using the formula:

$$\text{Diversity Score} = \frac{1}{\binom{n}{2}} \sum_{i=1}^{n-1} \sum_{j=i+1}^{n} d(S_i, S_j), \tag{20}$$

where $d(S_i, S_j)$ represents the edit distance between any two sequences $S_i$ and $S_j$. This calculation provides an empirical gauge of ProteinReDiff's ability to enrich the protein sequence space with novel and diverse sequences, underlining the practical variance introduced by our model.

### 5.2.3 Structure Preservation

Structural preservation is paramount in the redesign of proteins, ensuring that essential functional and structural characteristics are maintained post-modification. To effectively measure structural preservation between the original and redesigned proteins, three key metrics: the Template Modeling Score (TM Score) (Zhang & Skolnick, 2004), the Root Mean Square Deviation (RMSD) (Laskowski & de Beer, 2014), and the Contact Overlap (CO) (Bastolla et al., 2023). These two metrics collectively provide a comprehensive assessment of structural integrity and similarity, essential for evaluating the success of our protein redesign efforts.

**The Root Mean Square Deviation (RMSD)** is a measure used to quantify the distance between two sets of points. In the context of protein structures, these points are the positions of the atoms in the protein. The RMSD is given by the formula:

$$\text{RMSD}(\mathbf{p}, \mathbf{p}') = \min_{(R,t) \in \text{SO}(3) \times \mathbb{R}_3} \left[ \frac{1}{N} \sum_{i=1}^{N} \| p_i - (R p_i' + t) \|_2^2 \right]^{1/2}, \tag{21}$$

where $\mathbf{p} = (x_i, y_i, z_i)_{i=1}^{N}$ and $\mathbf{p}' = (x_i', y_i', z'i)_{i=1}^{N}$ denote two sequences of $N$ 3D coordinates representing the atomic positions in the original and redesigned proteins, respectively. This formula calculates the minimum root mean square of distances between corresponding atoms, after optimal superposition, which involves finding the best-fit rotation $R$ and translation $t$ that aligns the two sets of points. A lower RMSD value indicates a higher degree of structural similarity, making it a direct measure of the extent to which structural deviation has been minimized. Achieving a low RMSD is desirable, as it signifies that the redesign process has successfully preserved the core structural configuration of the original protein.

**TM Score** provides a normalized measure of structural similarity between protein configurations, which is less sensitive to local variations and more reflective of the overall topology. The TM Score is defined as follows:

$$\text{TM Score}(\mathbf{p}, \mathbf{p}') = \max_{(R,t) \in \text{SO}(3) \times \mathbb{R}_3} \left[ \frac{1}{1 + \frac{1}{N} \sum_{i=1}^{N} \frac{\| p_i - (R p_i' + t) \|_2^2}{d_0^2}} \right], \tag{22}$$

where $d_0$ is a scale parameter typically chosen based on the size of the proteins. The closer the TM Score is to 1, the more similar the structures are, indicating global structural alignment.

**Contact Overlap (CO)** provides a complementary perspective to RMSD and TM Score by focusing on the preservation of local structural motifs rather than overall geometric similarity. Several studies show that having high CO indicates protein's residue pairs having co-evolutionary signals (Cheng et al., 2019; Bastolla et al., 2023) and performing related functions (Iyer et al., 2020). CO quantitatively measures the conservation of inter-atomic contacts between the original and redesigned protein structures, which are crucial for the protein's structural integrity and functional capabilities. The metric is defined as:

$$\text{CO}(\mathbf{p}, \mathbf{p}') = \frac{|C \cap C'|}{|C \cup C'|}, \tag{23}$$

where $C = \{(i,j) : \| p_i - p_j \| < r_c, i \neq j\}$ and $C' = \{(i,j) : \| p_i' - p_j' \| < r_c, i \neq j\}$ represent the sets of contacts in the original and redesigned proteins, respectively. Here, $p_i$ and $p_i'$ are the positions of atoms in the original and redesigned proteins, and $r_c$ is a predefined cutoff distance that determines when two atoms are considered to be in contact. A high CO score indicates that many of the original contacts are preserved in the redesigned structure, suggesting that the redesign maintains much of the original protein's structural network, which is crucial for its stability and function.

### 5.2.4 Experimental Setup

To evaluate ProteinReDiff, we employed Omegafold (Wu et al., 2022b) to predict the three-dimensional structures of all designed protein sequences. The choice of Omegafold over AF2 was favorable because

Omegafold can more accurately fold proteins with low similarity to existing proteomes, making it suitable for proteins lacking available ligand-binding conformations. Next, we utilized AutoDock Vina (Trott & Olson, 2010) to conduct docking simulations and evaluate the binding affinity between the redesigned proteins and their respective ligands based on the predicted 3D structures. To ensure fair comparisons and mitigate potential biases introduced by pre-docked structures, we aligned our redesigned protein structures with reference structures before docking. This approach is crucial, particularly because the use of pre-docked structures may favor certain conformations, leading to inaccurate evaluations. Additionally, to provide context for our results, we compared the binding scores of our redesigned proteins not only with those of the original proteins but also with proteins generated by other protein design models. Although these models may exhibit different sequence characteristics compared to those explicitly designed for ligand binding affinity, comparing their scores offers valuable insights. Such comparisons help elucidate the interplay between protein sequence and structure in determining ligand interactions, enriching the interpretation of our findings and advancing our understanding of protein-ligand interactions.

**Benchmark Model Selection**  In selecting benchmark models for performance comparison, we focused on state-of-the-art approaches, particularly those relevant to protein design tasks. Traditionally, protein design has been primarily based on inverse folding, utilizing protein structure information. Our choices encompass a range of methodologies:

- MIF (Yang et al., 2022), MIF-ST (Yang et al., 2022), and ProteinMPNN (Dauparas et al., 2022) are notable for generating sequences with high identity and experimental significance, utilizing protein structure information.

- The Protein Generator (Lisanza et al., 2023), a representative of RosettaFold models (Watson et al., 2023), employs diffusion-based methods, making it an intriguing comparative candidate. The model also shares a similar loss function, $L_{KL}$, in sequence space with our model but diverges in modules and training procedures (i.e., stochastic masking).

- ESMIF (Hsu et al., 2022), belonging to the ESM model family (Lin et al., 2023b), stands as another competitive benchmark, emphasizing the generation of high-quality sequences.

- CARP, while lacking ligand information, shares similar protein input and output characteristics with our models, warranting inclusion for comparison.

- DPL (Nakata et al., 2023), originally geared towards protein-ligand complex generation, was adapted for our purposes by modifying loss functions and incorporating a sequence prediction module, given its alignment with our model architecture.

- LigandMPNN (Dauparas et al., 2023), resembling the most to our task in designing ligand-binding proteins, necessitates binding pocket information, unlike our model, which emphasizes a simplified yet effective approach for ligand-binding protein tasks.

Our model's design prioritizes simplicity in input while achieving effectiveness in output for ligand-binding protein tasks. For a comprehensive comparison of input-output dynamics across each model, please consult Table 3.

### 5.2.5  Results and Discussion

We conducted comprehensive evaluation of ProteinReDiff, as detailed in Table 4 and visually represented in Figure 6, across the metrics of ligand binding affinity, sequence diversity, and structure preservation. These evaluations provide a clear depiction of the model's performance relative to established baselines and within its variations.

For ProteinReDiff, we aimed to capture the diverse conformations of ligand-binding proteins, recognizing that they can adopt multiple structural states. To assess these conformations, we employed alignment metrics such as TM score, RMSD, and contact overlap (CO). In Figure 5, we presented several instances

Table 3: Comparison of protein design models based on input and output characteristics

| Model | Input | | | | Output | | |
|---|---|---|---|---|---|---|---|
| | Protein Sequence | Protein Structure | Ligand SMILES | Binding Pocket | Protein Sequence | Protein Structure | Ligand Structure |
| CARP (Yang et al., 2023a) | ✓ | × | × | × | ✓ | × | × |
| ESMIF (Hsu et al., 2022) | × | ✓ | × | × | ✓ | × | × |
| MIF (Yang et al., 2022) | ✓ | ✓ | × | × | ✓ | × | × |
| MIF-ST (Yang et al., 2022) | ✓ | ✓ | × | × | ✓ | × | × |
| ProteinMPNN (Dauparas et al., 2022) | × | ✓ | × | × | ✓ | × | × |
| LigandMPNN (Dauparas et al., 2023) | × | ✓ | ✓ | ✓ | ✓ | × | × |
| Protein Generator (Lisanza et al., 2023) | × | ✓ | × | × | ✓ | × | × |
| DPL (Nakata et al., 2023) | ✓ | × | ✓ | × | × | ✓ | ✓ |
| ProteinReDiff (Ours) | ✓ | × | ✓ | × | ✓ | ✓ | ✓ |

where the contact overlap appeared to be maintained, yet the RMSD is large and TM score is low. This discrepancy suggests that while global alignment metrics like TM score and RMSD may not adequately capture the domain shift within these complex ensembles, the preservation of local motifs, as indicated by contact overlap, remains crucial in our framework. This underscores the importance of capturing both global and local structural features for a comprehensive understanding of protein-ligand interactions.

A pivotal observation from our study is ProteinReDiff's unparalleled ability to enhance ligand binding affinity, particularly at a 15% masking ratio in Figure 6. This configuration not only surpasses the performance of Inverse Folding (IF) models and the original DPL framework but also exceeds the binding efficiencies of the original protein designs. By incorporating attention modules from AlphaFold2, ProteinReDiff effectively captures the complex interplay between proteins and ligands, demonstrating its superiority over the original DPL model. While other masking ratios within ProteinReDiff show varying degrees of effectiveness, lower ratios, though at the same par as reference, do not achieve the peak LBA performance observed at 15%. For instance, the 5% masked model emphasizes structural consistency with a high TM-Score and low RMSD, but does not exhibit the same level of binding capability as the 15% masking. These findings are also consistent with ablation studies shown in Appendix C.1. Conversely, higher masking ratios fail to strike the necessary balance between introducing beneficial modifications and maintaining functional precision, underscoring the importance of optimizing the masking ratio.

Our analysis of sequence diversity and structure preservation metrics reveals a delicate balance essential in protein redesign. The 15% masking ratio, identified as optimal for enhancing ligand binding affinity in our model, also aligns closely with benchmark methods in both sequence diversity and structure preservation. For instance, LigandMPNN excels in sequence diversity but faces challenges in obtaining binding pocket inputs for various design tasks, unlike our approach. Moreover, our models (at 30% and 40% maskings) significantly outperform others in contact overlap, crucial for diversifying structures while preserving functional motifs in protein redesign tasks. This equilibrium underscores ProteinReDiff's ability to optimize ligand interactions without compromising the exploration of sequence diversity or the integrity of original protein structures.

In contrast, extreme values in either sequence diversity or structure preservation, which could be seen in other masking ratios, do not lead to optimal ligand binding affinities. This finding highlights an inverse relationship between pushing the limits of diversity and preservation and achieving the primary goal of binding enhancement. Thus, the 15% masking ratio not only stands out for its ability to significantly improve ligand binding affinity but also for maintaining a balanced approach, ensuring that enhancements in functionality do not detract from the protein's structural and functional viability.

In Figure 7, we compare the ligand-binding affinity (LBA) of original and redesigned proteins by ProteinReDiff. The redesigned proteins maintain their original folds while significantly enhancing LBA. In ablation studies (Section 5.2.6), we can apply various masking strategies to adjust both sequence diversity and structural integrity. This approach has potential applications in different settings to control the affinity of ligand binders.

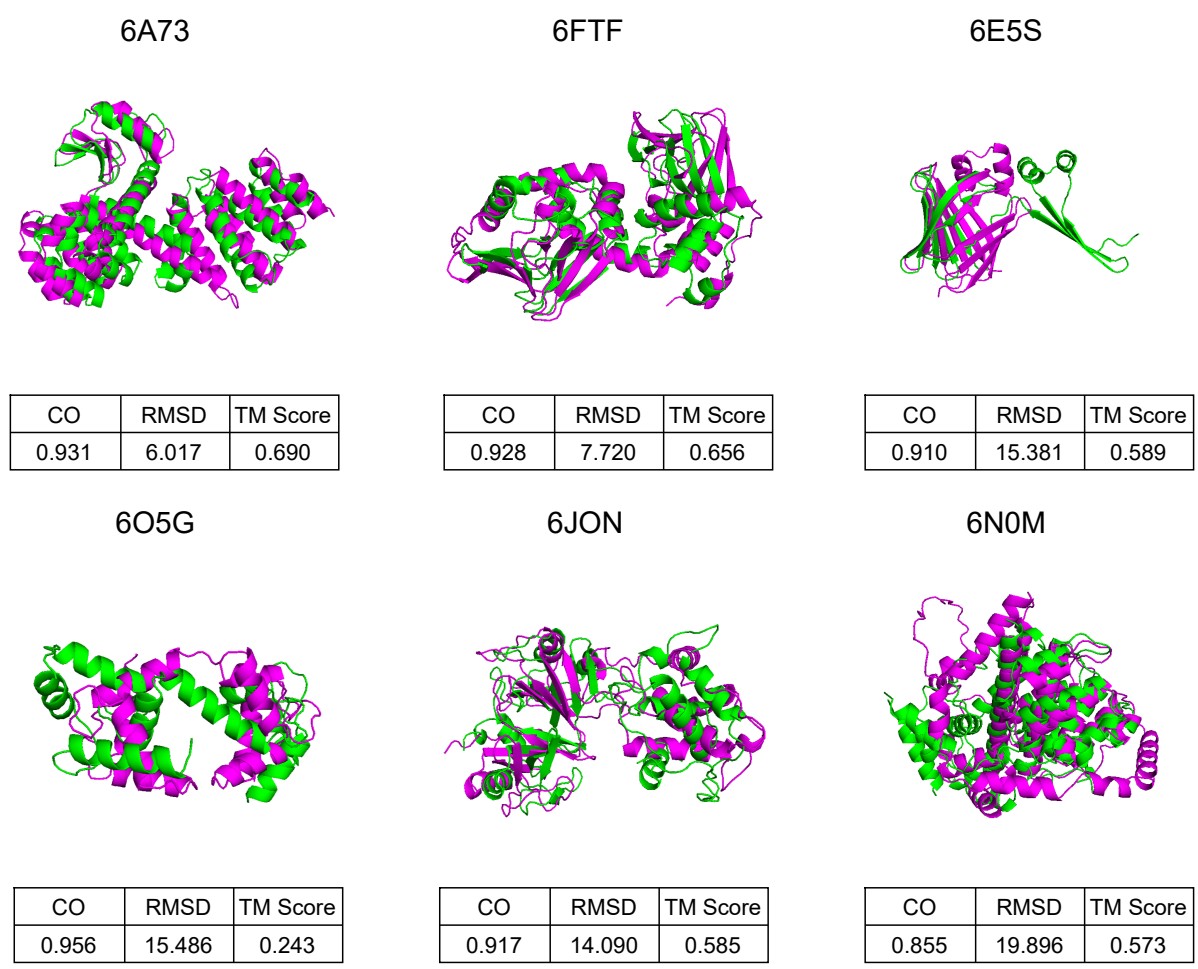

Figure 5: Comparative visualizations of protein structures, each annotated with its corresponding PDB ID. The figure includes a succinct table detailing Contact Overlap (CO) and Root Mean Square Deviation (RMSD) metrics. Original protein structures are highlighted in green, and the redesigned versions by ProteinReDiff are depicted in pink, illustrating the precise structural changes and enhancements achieved through the redesign.

### 5.2.6 Ablation Studies

Here we conducted thorough ablation studies on ProteinReDiff's model architecture, featurization, and masking ratios. For complete ablation setup, please refer to Table 7 (Appendix C.2)

**Interpreting Model Architecture** We trained ablated versions of ProteinReDiff without the SRA or OPU modules and compared them to the original DPL model. Initially designed for generating ensembles of complex structures, DPL was adapted for targeted protein redesign by adding sequence-based loss functions to generate new target sequences.

In Figure 8, we computed the performance score by averaging the sum of five evaluation metrics introduced in Sections 5.2.1, 5.2.2, and 5.2.3. Since the sequence diversity is not within the [0,1] range, we applied Min-Max normalization. For LBA and RMSD, we used inverse normalization to ensure that a score closer to 1.0 indicates better model performance. The average score is then compared with the baseline score of ProteinReDiff which was trained without any ablations.

We observed that our model outperformed DPL by a large margin. Incorporating just the OPU module (without the SRA module) yields better performance than DPL, indicating OPU's ability to exchange

Table 4: Comparison of method performance across multiple metrics: Ligand binding affinity (LBA), sequence diversity, and structure preservation. Ligand binding affinity (LBA), TM Score, and RMSD are reported as mean values with their respective margins of error.

| Category | Method | LBA (kcal/mol) ↓ | Sequence diversity ↑ | Structure preservation | | |
|---|---|---|---|---|---|---|
| | | | | TM Score ↑ | RMSD (Å) ↓ | CO ↑ |
| Baseline | CARP | -5.658 ± 0.301 | 185.532 | 0.850 ± 0.023 | 3.768 ± 0.553 | 0.922 ± 0.003 |
| | MIF | -5.518 ± 0.381 | 185.600 | **0.877** ± 0.020 | **2.986** ± 0.468 | 0.938 ± 0.002 |
| | MIF-ST | -5.596 ± 0.330 | 185.584 | 0.872 ± 0.021 | 3.026 ± 0.451 | 0.937 ± 0.003 |
| | ESMIF | -5.555 ± 0.326 | 187.512 | 0.837 ± 0.021 | 4.000 ± 0.501 | 0.915 ± 0.003 |
| | ProteinMPNN | -5.423 ± 0.225 | 188.792 | 0.714 ± 0.026 | 6.806 ± 0.616 | 0.859 ± 0.004 |
| | LigandMPNN | -5.717 ± 0.287 | **191.384** | 0.782 ± 0.024 | 4.512 ± 0.668 | 0.915 ± 0.008 |
| | Protein Generator | -5.674 ± 0.266 | 186.962 | 0.806 ± 0.022 | 4.431 ± 0.523 | 0.899 ± 0.003 |
| | DPL | -5.551 ± 0.459 | 188.139 | 0.788 ± 0.024 | 5.094 ± 0.537 | 0.896 ± 0.009 |
| | Reference cases | -5.847 ± 0.263 | - | - | - | - |
| ProteinReDiff (Ours) | 5% Masking | -5.805 ± 0.252 | 185.935 | 0.864 ± 0.022 | 3.197 ± 0.470 | **0.942** ± 0.007 |
| | 15% Masking | **-6.803** ± 0.329 | 186.627 | 0.845 ± 0.023 | 3.690 ± 0.508 | 0.935 ± 0.007 |
| | 30% Masking | -5.769 ± 0.244 | 187.877 | 0.803 ± 0.024 | 4.467 ± 0.544 | 0.916 ± 0.008 |
| | 40% Masking | -5.617 ± 0.366 | 188.600 | 0.756 ± 0.026 | 5.639 ± 0.625 | 0.896 ± 0.008 |
| | 60% Masking | -5.467 ± 0.318 | 190.425 | 0.305 ± 0.024 | 18.056 ± 0.773 | 0.735 ± 0.010 |
| | 70% Masking | -5.470 ± 0.199 | 187.291 | 0.147 ± 0.004 | 23.197 ± 0.497 | 0.689 ± 0.007 |

insights between single and pair representations. Firstly, the equivariant loss function is parameterized on the structural space, making the pairwise representations from the OPU critical to that loss. Secondly, without OPU, the model performs poorly on TMScore (the bottom brown line in Figure 11, Appendix C.2), which measures global structural preservation. Additionally, introducing SRA only without OPU hurts our model performance, suggesting the model would have been over-parameterized as the SRA updates primarily on the sequence representation. Therefore, combining both the OPU and SRA modules provides an effective approach for enhancing the representational learning of ProteinReDiff. A complete comparative assessment is presented in Table 4 and Appendix C.2.

**Ablations on Input Featurization Methods** We conducted ablation studies to evaluate different input featurization methods, including manual feature engineering for ligands and the use of ESM-2 as a pre-trained LLM for protein featurization.

We gradually reduced ligand features, starting with ligand distance and bond information (e.g., types, ring), and even omitted the entire bond and ligand. In Figure 8, omitting bond features and distance caused less reduction in model performance than omitting the entire ligand. Ligand bond information is crucial for the model to learn the relative positions of ligand atoms and adhere to geometric constraints within the triangular update module (Section 4.2.3).

We observed a significant decrease in model performance when ESM embeddings were excluded (the red bar in Figure 8). The ESM features alone (the brown bar) significantly boosted performance when training without ligand data, as these embeddings are enriched with protein evolutionary and biophysical information needed for both single and pair representations. Other protein features, such as position encodings and amino acid types, provided slight improvements, though they were minimal. However, excluding ligand information led to a reduction in model performance compared to the baseline, as the model relies on learning the overall structure of the complexes.

Therefore, using pre-trained featurization methods, such as ESM and other protein BERT-like models, in combination with ligand input, significantly enhances model training and performance.

**Impact of Masking Ratios** We examined ProteinReDiff's performance with various percentages of masked amino acids, adjusting the masking ratio as a hyperparameter and retraining our model. In Figure 9, we observed consistent top performance across the metrics with masking ratios between 5% and 15%. This

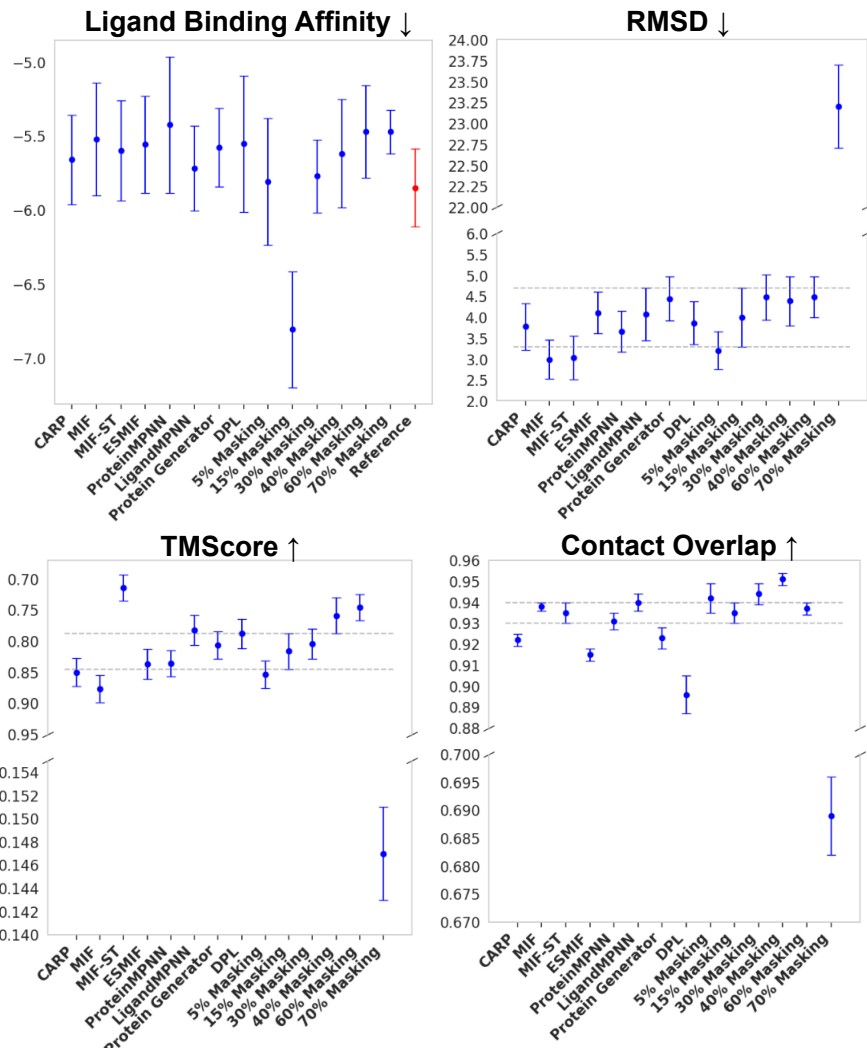

Figure 6: Visualization of method performance across metrics. The metrics are plotted with mean values and margins of error. For LBA, the red bar (top right) shows the docking score of reference complexes. The horizontal dash lines indicate the regions of 15% masking model which is our standard for comparison.

range is crucial for the protein redesign strategy, enhancing binding affinity while preserving the structural and functional motifs of the target protein. The 15% masking ratio achieved the best ligand binding affinity, the most important metric for capturing protein function.

Interestingly, we noticed performance spikes for 50% masking in contact overlap and TM-score. This is because applying stochastic masks allows the model to learn representations with varied masking from 0 up to the set ratio. Although the 50% masking does not surpass the 15% masking's performance, the improvement in the high masking regime demonstrates the robustness of our training scheme.

Overall, this investigation highlights the optimal level of sequence masking needed to enhance ligand binding affinity, sequence diversity, and structural preservation. It also reinforces training strategies for protein redesign as shown on the Discussion section (5.2.5).

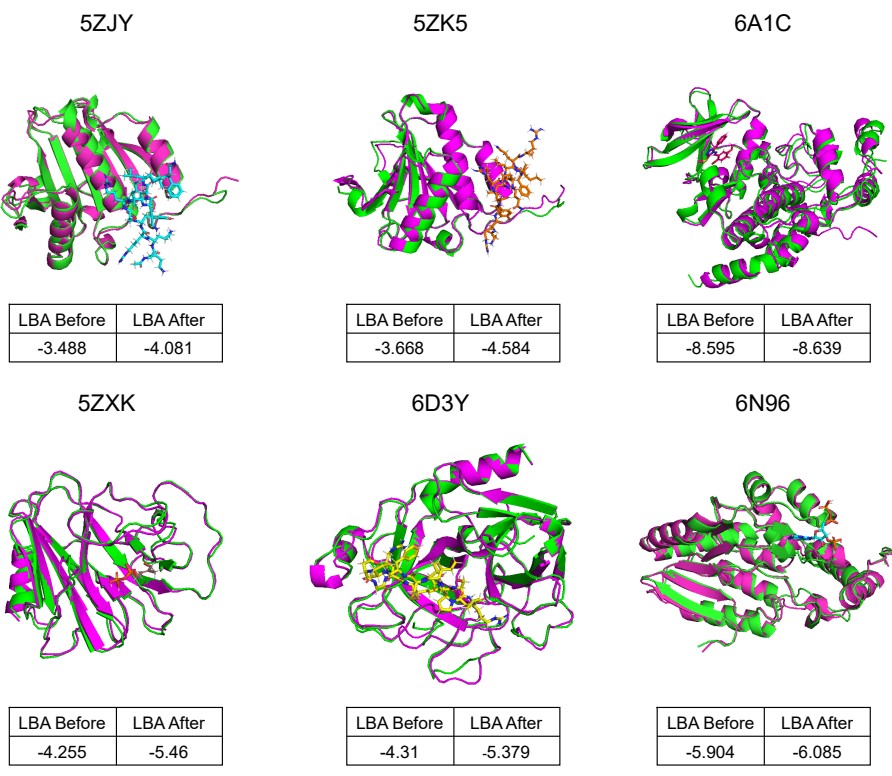

Figure 7: Comparative visualizations of protein-ligand complexes, each labeled with corresponding PDB IDs and accompanied by a small table showing Ligand Binding Affinity (LBA) before and after the redesign. Original structures are highlighted in green, while redesigned versions by ProteinReDiff appear in pink. Ligands are depicted in various colors to emphasize specific binding sites and molecular interaction enhancements post-redesign.

## 6 Conclusions

This study introduces ProteinReDiff, a computational framework developed to redesign ligand-binding proteins. By utilizing advanced techniques inspired by Equivariant Diffusion-Based Generative Models and the attention mechanism from AlphaFold2, ProteinReDiff demonstrates its ability to enhance complex protein-ligand interactions. Our model excels in optimizing ligand binding affinity based solely on initial protein sequences and ligand SMILES strings, bypassing the need for detailed structural data. Experimental validations highlight ProteinReDiff's capability to improve ligand binding affinity while preserving essential sequence diversity and structural integrity. These findings open new possibilities for protein-ligand complex modeling, indicating significant potential for ProteinReDiff in various biotechnological and pharmaceutical applications.

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

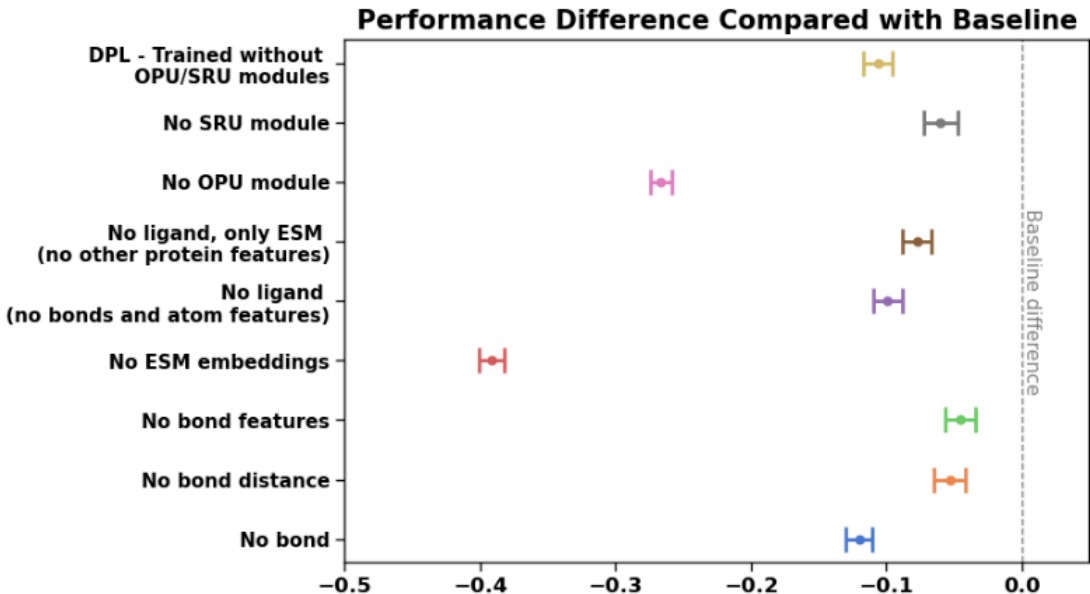

Figure 8: Ablation studies on ProteinReDiff's model architecture and featurization. The dash line indicates the baseline's average score obtained from ProteinReDiff without ablations.

Namrata Anand and Possu Huang. Generative modeling for protein structures. In S. Bengio, H. Wallach, H. Larochelle, K. Grauman, N. Cesa-Bianchi, and R. Garnett (eds.), *Advances in Neural Information Processing Systems*, volume 31. Curran Associates, Inc., 2018. URL `https://proceedings.neurips.cc/paper_files/paper/2018/file/afa299a4d1d8c52e75dd8a24c3ce534f-Paper.pdf`.

Frances H Arnold and Alexander A Volkov. Directed evolution of biocatalysts. *Current Opinion in Chemical Biology*, 3(1):54–59, 1999. ISSN 1367-5931. doi: https://doi.org/10.1016/S1367-5931(99)80010-6. URL `https://www.sciencedirect.com/science/article/pii/S1367593199800106`.

Peizhen Bai, Filip Miljković, Bino John, and Haiping Lu. Interpretable bilinear attention network with domain adaptation improves drug–target prediction. *Nature Machine Intelligence*, 5(2):126–136, Feb 2023a. ISSN 2522-5839. doi: 10.1038/s42256-022-00605-1. URL `https://doi.org/10.1038/s42256-022-00605-1`.

Peizhen Bai, Filip Miljković, Bino John, and Haiping Lu. Interpretable bilinear attention network with domain adaptation improves drug–target prediction. *Nature Machine Intelligence*, 5(2):126–136, Feb 2023b. ISSN 2522-5839. doi: 10.1038/s42256-022-00605-1. URL `https://doi.org/10.1038/s42256-022-00605-1`.

Emilia P. Barros, Jamie M. Schiffer, Anastassia Vorobieva, Jiayi Dou, David Baker, and Rommie E. Amaro. Improving the efficiency of ligand-binding protein design with molecular dynamics simulations. *Journal of Chemical Theory and Computation*, 15(10):5703–5715, Aug 2019. doi: 10.1021/acs.jctc.9b00483.

Ugo Bastolla, David Abia, and Oscar Piette. PC_ali: a tool for improved multiple alignments and evolutionary inference based on a hybrid protein sequence and structure similarity score. *Bioinformatics*, 39(11):btad630, 10 2023. ISSN 1367-4811. doi: 10.1093/bioinformatics/btad630. URL `https://doi.org/10.1093/bioinformatics/btad630`.

Nathaniel R. Bennett, Joseph L. Watson, Robert J. Ragotte, Andrew J. Borst, Déjenaé L. See, Connor Weidle, Riti Biswas, Ellen L. Shrock, Philip J. Y. Leung, Buwei Huang, Inna Goreshnik, Russell Ault, Kenneth D. Carr, Benedikt Singer, Cameron Criswell, Dionne Vafeados, Mariana Garcia Sanchez, Ho Min Kim, Susana Vázquez Torres, Sidney Chan, and David Baker. Atomically accurate de novo design of

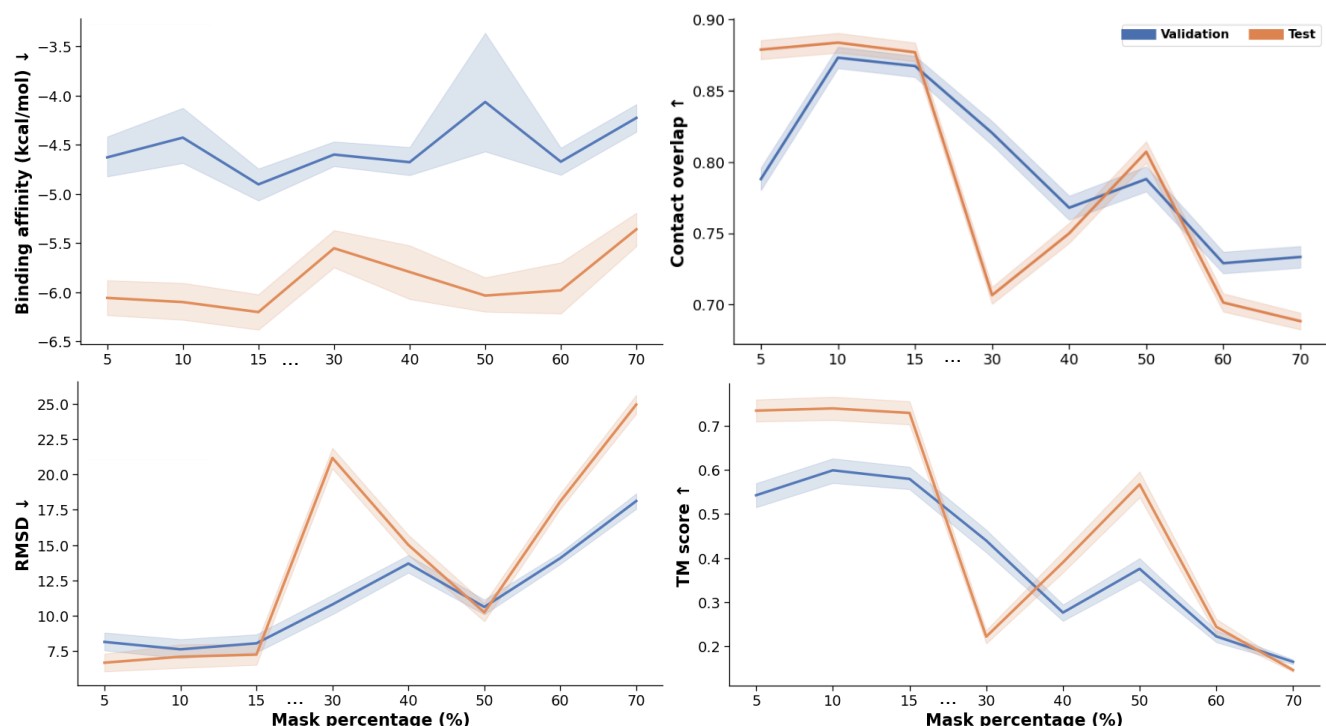

Figure 9: Mask ablation studies on both validation and test sets. Each of the mask ratios (5%, 10%, 15%, 30%, 40%, 50%, 60%, 70%) is a hyperparameter and represented by a model. The performances of the masked models are evaluated for all metrics. The arrows on y-axes show directions of better performance.

single-domain antibodies. *bioRxiv*, 2024. doi: 10.1101/2024.03.14.585103. URL `https://www.biorxiv.org/content/early/2024/03/18/2024.03.14.585103`.

Nadav Brandes, Dan Ofer, Yam Peleg, Nadav Rappoport, and Michal Linial. ProteinBERT: a universal deep-learning model of protein sequence and function. *Bioinformatics*, 38(8):2102–2110, 02 2022. ISSN 1367-4803. doi: 10.1093/bioinformatics/btac020. URL `https://doi.org/10.1093/bioinformatics/btac020`.

David Brookes, Hahnbeom Park, and Jennifer Listgarten. Conditioning by adaptive sampling for robust design. In Kamalika Chaudhuri and Ruslan Salakhutdinov (eds.), *Proceedings of the 36th International Conference on Machine Learning*, volume 97 of *Proceedings of Machine Learning Research*, pp. 773–782. PMLR, 09–15 Jun 2019. URL `https://proceedings.mlr.press/v97/brookes19a.html`.

Nicolas Buton, François Coste, and Yann Le Cunff. Predicting enzymatic function of protein sequences with attention. *Bioinformatics*, 39(10), October 2023.

Catherine Chen, Qihong Lu, Andre Beukers, Christopher Baldassano, and Kenneth A Norman. Learning to perform role-filler binding with schematic knowledge. *PeerJ*, 9(e11046):e11046, March 2021.

Lifan Chen, Xiaoqin Tan, Dingyan Wang, Feisheng Zhong, Xiaohong Liu, Tianbiao Yang, Xiaomin Luo, Kaixian Chen, Hualiang Jiang, and Mingyue Zheng. TransformerCPI: improving compound–protein interaction prediction by sequence-based deep learning with self-attention mechanism and label reversal experiments. *Bioinformatics*, 36(16):4406–4414, 05 2020. ISSN 1367-4803. doi: 10.1093/bioinformatics/btaa524. URL `https://doi.org/10.1093/bioinformatics/btaa524`.

Lixin Cheng, Pengfei Liu, Dong Wang, and Kwong-Sak Leung. Exploiting locational and topological overlap model to identify modules in protein interaction networks. *BMC Bioinformatics*, 20(1):23, January 2019.

Michael F. Chungyoun and Jeffrey J. Gray. Ai models for protein design are driving antibody engineering. *Current Opinion in Biomedical Engineering*, 28:100473, 2023. ISSN 2468-4511. doi: https://doi.org/10.1016/j.cobme.2023.100473. URL `https://www.sciencedirect.com/science/article/pii/S2468451123000296`.

Robert Creutznacher, Thorben Maass, Barbora Veselkova, George Ssebyatika, Thomas Krey, Martin Empting, Norbert Tautz, Martin Frank, Knut Kölbel, Charlotte Uetrecht, and Thomas Peters. Nmr experiments provide insights into ligand-binding to the sars-cov-2 spike protein receptor-binding domain. *Journal of the American Chemical Society*, 144(29):13060–13065, Jul 2022. ISSN 0002-7863. doi: 10.1021/jacs.2c05603. URL `https://doi.org/10.1021/jacs.2c05603`.

J. Dauparas, I. Anishchenko, N. Bennett, H. Bai, R. J. Ragotte, L. F. Milles, B. I. M. Wicky, A. Courbet, R. J. de Haas, N. Bethel, P. J. Y. Leung, T. F. Huddy, S. Pellock, D. Tischer, F. Chan, B. Koepnick, H. Nguyen, A. Kang, B. Sankaran, A. K. Bera, N. P. King, and D. Baker. Robust deep learning–based protein sequence design using proteinmpnn. *Science*, 378(6615):49–56, 2022. doi: 10.1126/science.add2187. URL `https://www.science.org/doi/abs/10.1126/science.add2187`.

Justas Dauparas, Gyu Rie Lee, Robert Pecoraro, Linna An, Ivan Anishchenko, Cameron Glasscock, and D. Baker. Atomic context-conditioned protein sequence design using ligandmpnn. *bioRxiv*, 2023. doi: 10.1101/2023.12.22.573103. URL `https://www.biorxiv.org/content/early/2023/12/23/2023.12.22.573103`.

Xing Du, Yi Li, Yuan-Ling Xia, Shi-Meng Ai, Jing Liang, Peng Sang, Xing-Lai Ji, and Shu-Qun Liu. Insights into protein-ligand interactions: Mechanisms, models, and methods. *Int. J. Mol. Sci.*, 17(2):144, January 2016.

Sasha B. Ebrahimi and Devleena Samanta. Engineering protein-based therapeutics through structural and chemical design. *Nature Communications*, 14(1), Apr 2023. doi: 10.1038/s41467-023-38039-x.

Ahmed Elnaggar, Michael Heinzinger, Christian Dallago, Ghalia Rehawi, Yu Wang, Llion Jones, Tom Gibbs, Tamas Feher, Christoph Angerer, Martin Steinegger, Debsindhu Bhowmik, and Burkhard Rost. Prottrans: Toward understanding the language of life through self-supervised learning. *IEEE Transactions on Pattern Analysis and Machine Intelligence*, 44(10):7112–7127, 2022. doi: 10.1109/TPAMI.2021.3095381.

Clara Fannjiang, Stephen Bates, Anastasios N. Angelopoulos, Jennifer Listgarten, and Michael I. Jordan. Conformal prediction under feedback covariate shift for biomolecular design. *Proceedings of the National Academy of Sciences*, 119(43), October 2022. ISSN 1091-6490. doi: 10.1073/pnas.2204569119. URL `http://dx.doi.org/10.1073/pnas.2204569119`.

Cong Fu, Keqiang Yan, Limei Wang, Wing Yee Au, Michael McThrow, Tao Komikado, Koji Maruhashi, Kanji Uchino, Xiaoning Qian, and Shuiwang Ji. A latent diffusion model for protein structure generation, 2023.

Luca Gagliardi and Walter Rocchia. SiteFerret: Beyond simple pocket identification in proteins. *J. Chem. Theory Comput.*, 19(15):5242–5259, August 2023.

Joe G. Greener, Lewis Moffat, and David T. Jones. Design of metalloproteins and novel protein folds using variational autoencoders. *Scientific Reports*, 8(1):16189, Nov 2018. ISSN 2045-2322. doi: 10.1038/s41598-018-34533-1. URL `https://doi.org/10.1038/s41598-018-34533-1`.

Nate Gruver, Samuel Stanton, Nathan C. Frey, Tim G. J. Rudner, Isidro Hotzel, Julien Lafrance-Vanasse, Arvind Rajpal, Kyunghyun Cho, and Andrew Gordon Wilson. Protein design with guided discrete diffusion, 2023.

Gurkan Guntas, Thomas J. Mansell, Jin Ryoun Kim, and Marc Ostermeier. Directed evolution of protein switches and their application to the creation of ligand-binding proteins. *Proceedings of the National Academy of Sciences*, 102(32):11224–11229, 2005. doi: 10.1073/pnas.0502673102. URL `https://www.pnas.org/doi/abs/10.1073/pnas.0502673102`.

Emiel Hoogeboom, Victor Garcia Satorras, Clément Vignac, and Max Welling. Equivariant diffusion for molecule generation in 3d, 2022.

Chloe Hsu, Robert Verkuil, Jason Liu, Zeming Lin, Brian Hie, Tom Sercu, Adam Lerer, and Alexander Rives. Learning inverse folding from millions of predicted structures. *ICML*, 2022. doi: 10.1101/2022.04. 10.487779. URL https://www.biorxiv.org/content/early/2022/04/10/2022.04.10.487779.

Kexin Huang, Cao Xiao, Lucas M Glass, and Jimeng Sun. MolTrans: Molecular Interaction Transformer for drug–target interaction prediction. *Bioinformatics*, 37(6):830–836, 10 2020. ISSN 1367-4803. doi: 10.1093/bioinformatics/btaa880. URL https://doi.org/10.1093/bioinformatics/btaa880.

Qiuyuan Huang, Paul Smolensky, Xiaodong He, Li Deng, and Dapeng Wu. Tensor product generation networks for deep NLP modeling. In Marilyn Walker, Heng Ji, and Amanda Stent (eds.), *Proceedings of the 2018 Conference of the North American Chapter of the Association for Computational Linguistics: Human Language Technologies, Volume 1 (Long Papers)*, pp. 1263–1273, New Orleans, Louisiana, June 2018. Association for Computational Linguistics. doi: 10.18653/v1/N18-1114. URL https://aclanthology.org/N18-1114.

Qiuyuan Huang, Li Deng, Dapeng Wu, Chang Liu, and Xiaodong He. Attentive tensor product learning. *Proceedings of the AAAI Conference on Artificial Intelligence*, 33(01):1344–1351, Jul. 2019. doi: 10.1609/aaai.v33i01.33011344. URL https://ojs.aaai.org/index.php/AAAI/article/view/3934.

John Ingraham, Vikas Garg, Regina Barzilay, and Tommi Jaakkola. Generative models for graph-based protein design. In H. Wallach, H. Larochelle, A. Beygelzimer, F. d'Alché-Buc, E. Fox, and R. Garnett (eds.), *Advances in Neural Information Processing Systems*, volume 32. Curran Associates, Inc., 2019. URL https://proceedings.neurips.cc/paper_files/paper/2019/file/f3a4ff4839c56a5f460c88cce3666a2b-Paper.pdf.

Shahid Iqbal, Fang Ge, Fuyi Li, Tatsuya Akutsu, Yuanting Zheng, Robin B. Gasser, Dong-Jun Yu, Geoffrey I. Webb, and Jiangning Song. Prost: Alphafold2-aware sequence-based predictor to estimate protein stability changes upon missense mutations. *Journal of Chemical Information and Modeling*, 62(17):4270–4282, Aug 2022. doi: 10.1021/acs.jcim.2c00799.

Touseef Iqbal and Shaima Qureshi. The survey: Text generation models in deep learning. *Journal of King Saud University - Computer and Information Sciences*, 34(6, Part A):2515–2528, 2022. ISSN 1319-1578. doi: https://doi.org/10.1016/j.jksuci.2020.04.001. URL https://www.sciencedirect.com/science/article/pii/S1319157820303360.

Mallika Iyer, Zhanwen Li, Lukasz Jaroszewski, Mayya Sedova, and Adam Godzik. Difference contact maps: From what to why in the analysis of the conformational flexibility of proteins. *PLoS One*, 15(3):e0226702, March 2020.

Dejun Jiang, Chang-Yu Hsieh, Zhenxing Wu, Yu Kang, Jike Wang, Ercheng Wang, Ben Liao, Chao Shen, Lei Xu, Jian Wu, Dongsheng Cao, and Tingjun Hou. Interactiongraphnet: A novel and efficient deep graph representation learning framework for accurate protein–ligand interaction predictions. *Journal of Medicinal Chemistry*, 64(24):18209–18232, 2021. doi: 10.1021/acs.jmedchem.1c01830. URL https://doi.org/10.1021/acs.jmedchem.1c01830. PMID: 34878785.

Mingjian Jiang, Zhen Li, Shugang Zhang, Shuang Wang, Xiaofeng Wang, Qing Yuan, and Zhiqiang Wei. Drug–target affinity prediction using graph neural network and contact maps. *RSC Adv.*, 10:20701–20712, 2020. doi: 10.1039/D0RA02297G. URL http://dx.doi.org/10.1039/D0RA02297G.

James M. Joyce. *Kullback-Leibler Divergence*, pp. 720–722. Springer Berlin Heidelberg, Berlin, Heidelberg, 2011. ISBN 978-3-642-04898-2. doi: 10.1007/978-3-642-04898-2_327. URL https://doi.org/10.1007/978-3-642-04898-2_327.

Fusong Ju, Jianwei Zhu, Bin Shao, Lupeng Kong, Tie-Yan Liu, Wei-Mou Zheng, and Dongbo Bu. CopulaNet: Learning residue co-evolution directly from multiple sequence alignment for protein structure prediction. *Nat. Commun.*, 12(1):2535, May 2021.

John Jumper, Richard Evans, Alexander Pritzel, Tim Green, Michael Figurnov, Olaf Ronneberger, Kathryn Tunyasuvunakool, Russ Bates, Augustin Žídek, Anna Potapenko, Alex Bridgland, Clemens Meyer, Simon A. A. Kohl, Andrew J. Ballard, Andrew Cowie, Bernardino Romera-Paredes, Stanislav Nikolov, Rishub Jain, Jonas Adler, Trevor Back, Stig Petersen, David Reiman, Ellen Clancy, Michal Zielinski, Martin Steinegger, Michalina Pacholska, Tamas Berghammer, Sebastian Bodenstein, David Silver, Oriol Vinyals, Andrew W. Senior, Koray Kavukcuoglu, Pushmeet Kohli, and Demis Hassabis. Highly accurate protein structure prediction with alphafold. *Nature*, 596(7873):583–589, Aug 2021. ISSN 1476-4687. doi: 10.1038/s41586-021-03819-2. URL `https://doi.org/10.1038/s41586-021-03819-2`.

Diederik P. Kingma, Tim Salimans, Ben Poole, and Jonathan Ho. Variational diffusion models, 2023.

David Ryan Koes, Matthew P. Baumgartner, and Carlos J. Camacho. Lessons learned in empirical scoring with smina from the csar 2011 benchmarking exercise. *Journal of Chemical Information and Modeling*, 53(8):1893–1904, Aug 2013. ISSN 1549-9596. doi: 10.1021/ci300604z. URL `https://doi.org/10.1021/ci300604z`.

Huan Yee Koh, Anh TN Nguyen, Shirui Pan, Lauren T May, and Geoffrey I Webb. Psichic: physicochemical graph neural network for learning protein-ligand interaction fingerprints from sequence data. *bioRxiv*, pp. 2023–09, 2023.

Ivan V Korendovych. Rational and semirational protein design. *Protein engineering: methods and protocols*, pp. 15–23, 2018.

Tim Kucera, Matteo Togninalli, and Laetitia Meng-Papaxanthos. Conditional generative modeling for de novo protein design with hierarchical functions. *Bioinformatics*, 38(13):3454–3461, 05 2022. ISSN 1367-4803. doi: 10.1093/bioinformatics/btac353. URL `https://doi.org/10.1093/bioinformatics/btac353`.

Roman Laskowski and Tjaart de Beer. *Root Mean Square Deviation (RMSD)*. John Wiley and Sons, Ltd, 2014. ISBN 9780471650126. doi: https://doi.org/10.1002/9780471650126.dob0640.pub2. URL `https://onlinelibrary.wiley.com/doi/abs/10.1002/9780471650126.dob0640.pub2`.

Shuangli Li, Jingbo Zhou, Tong Xu, Liang Huang, Fan Wang, Haoyi Xiong, Weili Huang, Dejing Dou, and Hui Xiong. Structure-aware interactive graph neural networks for the prediction of protein-ligand binding affinity. In *Proceedings of the 27th ACM SIGKDD Conference on Knowledge Discovery & Data Mining*, KDD '21, pp. 975–985, New York, NY, USA, 2021. Association for Computing Machinery. ISBN 9781450383325. doi: 10.1145/3447548.3467311. URL `https://doi.org/10.1145/3447548.3467311`.

Yeqing Lin and Mohammed AlQuraishi. Generating novel, designable, and diverse protein structures by equivariantly diffusing oriented residue clouds. In *Proceedings of the 40th International Conference on Machine Learning*, ICML'23. JMLR.org, 2023.

Zeming Lin, Halil Akin, Roshan Rao, Brian Hie, Zhongkai Zhu, Wenting Lu, Nikita Smetanin, Robert Verkuil, Ori Kabeli, Yaniv Shmueli, Allan dos Santos Costa, Maryam Fazel-Zarandi, Tom Sercu, Salvatore Candido, and Alexander Rives. Evolutionary-scale prediction of atomic-level protein structure with a language model. *Science*, 379(6637):1123–1130, 2023a. doi: 10.1126/science.ade2574. URL `https://www.science.org/doi/abs/10.1126/science.ade2574`.

Zeming Lin, Halil Akin, Roshan Rao, Brian Hie, Zhongkai Zhu, Wenting Lu, Nikita Smetanin, Robert Verkuil, Ori Kabeli, Yaniv Shmueli, Allan dos Santos Costa, Maryam Fazel-Zarandi, Tom Sercu, Salvatore Candido, and Alexander Rives. Evolutionary-scale prediction of atomic-level protein structure with a language model. *Science*, 379(6637):1123–1130, 2023b. doi: 10.1126/science.ade2574. URL `https://www.science.org/doi/abs/10.1126/science.ade2574`.

Sidney Lyayuga Lisanza, Jake Merle Gershon, Sam Tipps, Lucas Arnoldt, Samuel Hendel, Jeremiah Nelson Sims, Xinting Li, and David Baker. Joint generation of protein sequence and structure with rosettafold sequence space diffusion. *bioRxiv*, 2023. doi: 10.1101/2023.05.08.539766. URL `https://www.biorxiv.org/content/early/2023/05/10/2023.05.08.539766`.

Dina Listov, Casper A. Goverde, Bruno E. Correia, and Sarel Jacob Fleishman. Opportunities and challenges in design and optimization of protein function. *Nature Reviews Molecular Cell Biology*, Apr 2024. ISSN 1471-0080. doi: 10.1038/s41580-024-00718-y. URL https://doi.org/10.1038/s41580-024-00718-y.

Wei Lu, Qifeng Wu, Jixian Zhang, Jiahua Rao, Chengtao Li, and Shuangjia Zheng. Tankbind: Trigonometry-aware neural networks for drug-protein binding structure prediction. *bioRxiv*, 2022. doi: 10.1101/2022.06.06.495043. URL https://www.biorxiv.org/content/early/2022/06/06/2022.06.06.495043.

Meng Lv, Xufei Luo, Janne Estill, Yunlan Liu, Mengjuan Ren, Jianjian Wang, Qi Wang, Siya Zhao, Xiaohui Wang, Shu Yang, Xixi Feng, Weiguo Li, Enmei Liu, Xianzhuo Zhang, Ling Wang, Qi Zhou, Wenbo Meng, Xiaolong Qi, Yangqin Xun, Xuan Yu, Yaolong Chen, and COVID-19 evidence and recommendations working group. Coronavirus disease (COVID-19): a scoping review. *Euro Surveill.*, 25(15), April 2020.

Suyue Lyu, Shahin Sowlati-Hashjin, and Michael Garton. Proteinvae: Variational autoencoder for transla-tional protein design. *bioRxiv*, 2023. doi: 10.1101/2023.03.04.531110. URL https://www.biorxiv.org/content/early/2023/03/05/2023.03.04.531110.

Ali Madani, Ben Krause, Eric R. Greene, Subu Subramanian, Benjamin P. Mohr, James M. Holton, Jose Luis Olmos, Caiming Xiong, Zachary Z. Sun, Richard Socher, James S. Fraser, and Nikhil Naik. Large language models generate functional protein sequences across diverse families. *Nature Biotechnology*, 41(8):1099–1106, Aug 2023. ISSN 1546-1696. doi: 10.1038/s41587-022-01618-2. URL https://doi.org/10.1038/s41587-022-01618-2.

Andrew T. McNutt, Paul Francoeur, Rishal Aggarwal, Tomohide Masuda, Rocco Meli, Matthew Ragoza, Jocelyn Sunseri, and David Ryan Koes. Gnina 1.0: molecular docking with deep learning. *Journal of Cheminformatics*, 13(1):43, Jun 2021. ISSN 1758-2946. doi: 10.1186/s13321-021-00522-2. URL https://doi.org/10.1186/s13321-021-00522-2.

Artur Meller, Michael Ward, Jonathan Borowsky, Meghana Kshirsagar, Jeffrey M. Lotthammer, Felipe Oviedo, Juan Lavista Ferres, and Gregory R. Bowman. Predicting locations of cryptic pockets from single protein structures using the pocketminer graph neural network. *Nature Communications*, 14(1):1177, Mar 2023. ISSN 2041-1723. doi: 10.1038/s41467-023-36699-3. URL https://doi.org/10.1038/s41467-023-36699-3.

Frederic P. Miller, Agnes F. Vandome, and John McBrewster. *Levenshtein Distance: Information theory, Computer science, String (computer science), String metric, Damerau?Levenshtein distance, Spell checker, Hamming distance.* Alpha Press, 2009. ISBN 6130216904.

Xiaoping Min, Chongzhou Yang, Jun Xie, Yang Huang, Nan Liu, Xiaocheng Jin, Tianshu Wang, Zhibo Kong, Xiaoli Lu, Shengxiang Ge, Jun Zhang, and Ningshao Xia. Tpgen: a language model for stable protein design with a specific topology structure. *BMC Bioinformatics*, 25(1):35, Jan 2024. ISSN 1471-2105. doi: 10.1186/s12859-024-05637-5. URL https://doi.org/10.1186/s12859-024-05637-5.

Harry L. Morgan. The generation of a unique machine description for chemical structures-a technique developed at chemical abstracts service. *Journal of Chemical Documentation*, 5:107–113, 1965. URL https://api.semanticscholar.org/CorpusID:62164893.

Christian Munk, Kasper Harpsøe, Alexander S Hauser, Vignir Isberg, and David E Gloriam. Integrating structural and mutagenesis data to elucidate gpcr ligand binding. *Current Opinion in Pharmacology*, 30:51–58, Oct 2016. doi: 10.1016/j.coph.2016.07.003.

Shuya Nakata, Yoshiharu Mori, and Shigenori Tanaka. End-to-end protein–ligand complex structure gener-ation with diffusion-based generative models. *BMC Bioinformatics*, 24(1):233, Jun 2023. ISSN 1471-2105. doi: 10.1186/s12859-023-05354-5. URL https://doi.org/10.1186/s12859-023-05354-5.

Thin Nguyen, Hang Le, Thomas P Quinn, Tri Nguyen, Thuc Duy Le, and Svetha Venkatesh. GraphDTA: predicting drug–target binding affinity with graph neural networks. *Bioinformatics*, 37(8):1140–1147, 10 2020. ISSN 1367-4803. doi: 10.1093/bioinformatics/btaa921. URL https://doi.org/10.1093/bioinformatics/btaa921.

Vladimiras Oleinikovas, Giorgio Saladino, Benjamin P Cossins, and Francesco L Gervasio. Understanding cryptic pocket formation in protein targets by enhanced sampling simulations. *J. Am. Chem. Soc.*, 138 (43):14257–14263, November 2016.

Nicholas F. Polizzi and William F. DeGrado. A defined structural unit enables de novo design of small-molecule–binding proteins. *Science*, 369(6508):1227–1233, 2020. doi: 10.1126/science.abb8330. URL https://www.science.org/doi/abs/10.1126/science.abb8330.

Roshan M Rao, Jason Liu, Robert Verkuil, Joshua Meier, John Canny, Pieter Abbeel, Tom Sercu, and Alexander Rives. Msa transformer. In Marina Meila and Tong Zhang (eds.), *Proceedings of the 38th International Conference on Machine Learning*, volume 139 of *Proceedings of Machine Learning Research*, pp. 8844–8856. PMLR, 18–24 Jul 2021. URL https://proceedings.mlr.press/v139/rao21a.html.

Gaurav Raut and Apoorv Singh. Generative ai in vision: A survey on models, metrics and applications, 2024.

Alexander Rives, Joshua Meier, Tom Sercu, Siddharth Goyal, Zeming Lin, Jason Liu, Demi Guo, Myle Ott, C. Lawrence Zitnick, Jerry Ma, and Rob Fergus. Biological structure and function emerge from scaling unsupervised learning to 250 million protein sequences. *Proceedings of the National Academy of Sciences*, 118(15):e2016239118, 2021. doi: 10.1073/pnas.2016239118. URL https://www.pnas.org/doi/abs/10.1073/pnas.2016239118.

Jeffrey A. Ruffolo and Ali Madani. Designing proteins with language models. *Nature Biotechnology*, 42(2): 200–202, Feb 2024. ISSN 1546-1696. doi: 10.1038/s41587-024-02123-4. URL https://doi.org/10.1038/s41587-024-02123-4.

Annamaria Ruscito and Maria C. DeRosa. Small-molecule binding aptamers: Selection strategies, characterization, and applications. *Frontiers in Chemistry*, 4, May 2016. doi: 10.3389/fchem.2016.00014.

Ryusuke Sawada, Yuko Sakajiri, Tomokazu Shibata, and Yoshihiro Yamanishi. Predicting therapeutic and side effects from drug binding affinities to human proteome structures. *iScience*, 27(6):110032, June 2024.

Jeffrey M. Schaub, Chia-Wei Chou, Hung-Che Kuo, Kamyab Javanmardi, Ching-Lin Hsieh, Jory Goldsmith, Andrea M. DiVenere, Kevin C. Le, Daniel Wrapp, Patrick O. Byrne, and et al. Expression and characterization of sars-cov-2 spike proteins. *Nature Protocols*, 16(11):5339–5356, Oct 2021. doi: 10.1038/s41596-021-00623-0.

Imanol Schlag and Jürgen Schmidhuber. Learning to reason with third-order tensor products. November 2018.

Ian Sillitoe, Natalie Dawson, Tony E Lewis, Sayoni Das, Jonathan G Lees, Paul Ashford, Adeyelu Tolulope, Harry M Scholes, Ilya Senatorov, Andra Bujan, Fatima Ceballos Rodriguez-Conde, Benjamin Dowling, Janet Thornton, and Christine A Orengo. CATH: expanding the horizons of structure-based functional annotations for genome sequences. *Nucleic Acids Research*, 47(D1):D280–D284, 11 2018. ISSN 0305-1048. doi: 10.1093/nar/gky1097. URL https://doi.org/10.1093/nar/gky1097.

Jeffrey Skolnick and Hongyi Zhou. Implications of the essential role of small molecule ligand binding pockets in protein–protein interactions. *The Journal of Physical Chemistry B*, 126(36):6853–6867, Aug 2022. doi: 10.1021/acs.jpcb.2c04525.

Paul Smolensky. Tensor product variable binding and the representation of symbolic structures in connectionist systems. *Artificial Intelligence*, 46(1):159–216, 1990. ISSN 0004-3702. doi: https://doi.org/10.1016/0004-3702(90)90007-M. URL https://www.sciencedirect.com/science/article/pii/000437029090007M.

Yang Song, Jascha Sohl-Dickstein, Diederik P. Kingma, Abhishek Kumar, Stefano Ermon, and Ben Poole. Score-based generative modeling through stochastic differential equations. *arXiv*, 2020. doi: 10.48550/ARXIV.2011.13456. URL https://arxiv.org/abs/2011.13456.

Zhongdi Song, Qunfeng Zhang, Wenhui Wu, Zhongji Pu, and Haoran Yu. Rational design of enzyme activity and enantioselectivity. *Frontiers in Bioengineering and Biotechnology*, 11, Jan 2023. doi: 10.3389/fbioe. 2023.1129149.

Martin Steinegger and Johannes Söding. Mmseqs2 enables sensitive protein sequence searching for the analysis of massive data sets. *Nature Biotechnology*, 35(11):1026–1028, October 2017. ISSN 1546-1696. doi: 10.1038/nbt.3988. URL http://dx.doi.org/10.1038/nbt.3988.

Marta M Stepniewska-Dziubinska, Piotr Zielenkiewicz, and Pawel Siedlecki. Development and evaluation of a deep learning model for protein–ligand binding affinity prediction. *Bioinformatics*, 34(21):3666–3674, 05 2018. ISSN 1367-4803. doi: 10.1093/bioinformatics/bty374. URL https://doi.org/10.1093/bioinformatics/bty374.

Hannes Stärk, Bowen Jing, Regina Barzilay, and Tommi Jaakkola. Harmonic self-conditioned flow matching for multi-ligand docking and binding site design, 2023.

Baris E. Suzek, Yuqi Wang, Hongzhan Huang, Peter B. McGarvey, Cathy H. Wu, and the UniProt Consortium. UniRef clusters: a comprehensive and scalable alternative for improving sequence similarity searches. *Bioinformatics*, 31(6):926–932, 11 2014. ISSN 1367-4803. doi: 10.1093/bioinformatics/btu739. URL https://doi.org/10.1093/bioinformatics/btu739.

Freyr Sverrisson, Jean Feydy, Bruno E. Correia, and Michael M. Bronstein. Fast end-to-end learning on protein surfaces. In *2021 IEEE/CVF Conference on Computer Vision and Pattern Recognition (CVPR)*, pp. 15267–15276, 2021. doi: 10.1109/CVPR46437.2021.01502.

Diogo Tavares and Jan Roelof van der Meer. Ribose-binding protein mutants with improved interaction towards the non-natural ligand 1,3-cyclohexanediol. *Frontiers in Bioengineering and Biotechnology*, 9, Jul 2021. doi: 10.3389/fbioe.2021.705534.

Jeanne Trinquier, Guido Uguzzoni, Andrea Pagnani, Francesco Zamponi, and Martin Weigt. Efficient generative modeling of protein sequences using simple autoregressive models. *Nature Communications*, 12(1): 5800, Oct 2021. ISSN 2041-1723. doi: 10.1038/s41467-021-25756-4. URL https://doi.org/10.1038/s41467-021-25756-4.

Oleg Trott and Arthur J. Olson. Autodock vina: Improving the speed and accuracy of docking with a new scoring function, efficient optimization, and multithreading. *Journal of Computational Chemistry*, 31(2): 455–461, 2010. doi: https://doi.org/10.1002/jcc.21334. URL https://onlinelibrary.wiley.com/doi/abs/10.1002/jcc.21334.

Yann Waltenspühl, Jeliazko R. Jeliazkov, Lutz Kummer, and Andreas Plückthun. Directed evolution for high functional production and stability of a challenging g protein-coupled receptor. *Scientific Reports*, 11 (1), Apr 2021. doi: 10.1038/s41598-021-87793-9.

Karolina Wanat. Biological barriers, and the influence of protein binding on the passage of drugs across them. *Molecular Biology Reports*, 47(4):3221–3231, Mar 2020. doi: 10.1007/s11033-020-05361-2.

Meng Wang and Huimin Zhao. Combined and iterative use of computational design and directed evolution for protein–ligand binding design. *Methods in Molecular Biology*, pp. 139–153, 2016. doi: 10.1007/978-1-4939-3569-7_8.

Penglei Wang, Shuangjia Zheng, Yize Jiang, Chengtao Li, Junhong Liu, Chang Wen, Atanas Patronov, Dahong Qian, Hongming Chen, and Yuedong Yang. Structure-aware multimodal deep learning for drug–protein interaction prediction. *Journal of Chemical Information and Modeling*, 62(5):1308–1317, Mar 2022. ISSN 1549-9596. doi: 10.1021/acs.jcim.2c00060. URL https://doi.org/10.1021/acs.jcim.2c00060.

Renxiao Wang, Xueliang Fang, Yipin Lu, and Shaomeng Wang. The pdbbind database: Collection of binding affinities for protein-ligand complexes with known three-dimensional structures. *Journal of Medicinal Chemistry*, 47(12):2977–2980, 2004. doi: 10.1021/jm030580l. URL https://doi.org/10.1021/jm0305801. PMID: 15163179.

Joseph L. Watson, David Juergens, Nathaniel R. Bennett, Brian L. Trippe, Jason Yim, Helen E. Eisenach, Woody Ahern, Andrew J. Borst, Robert J. Ragotte, Lukas F. Milles, Basile I. M. Wicky, Nikita Hanikel, Samuel J. Pellock, Alexis Courbet, William Sheffler, Jue Wang, Preetham Venkatesh, Isaac Sappington, Susana Vázquez Torres, Anna Lauko, Valentin De Bortoli, Emile Mathieu, Sergey Ovchinnikov, Regina Barzilay, Tommi S. Jaakkola, Frank DiMaio, Minkyung Baek, and David Baker. De novo design of protein structure and function with rfdiffusion. *Nature*, 620(7976):1089–1100, Aug 2023. ISSN 1476-4687. doi: 10.1038/s41586-023-06415-8. URL `https://doi.org/10.1038/s41586-023-06415-8`.

David Weininger. Smiles, a chemical language and information system. 1. introduction to methodology and encoding rules. *Journal of Chemical Information and Computer Sciences*, 28(1):31–36, 1988. doi: 10.1021/ci00057a005. URL `https://doi.org/10.1021/ci00057a005`.

Kevin E. Wu, Kevin K. Yang, Rianne van den Berg, James Y. Zou, Alex X. Lu, and Ava P. Amini. Protein structure generation via folding diffusion, 2022a.

Ruidong Wu, Fan Ding, Rui Wang, Rui Shen, Xiwen Zhang, Shitong Luo, Chenpeng Su, Zuofan Wu, Qi Xie, Bonnie Berger, Jianzhu Ma, and Jian Peng. High-resolution de novo structure prediction from primary sequence. *bioRxiv*, 2022b. doi: 10.1101/2022.07.21.500999. URL `https://www.biorxiv.org/content/early/2022/07/22/2022.07.21.500999`.

Tian Xu, Qin Xu, and Jianyong Li. Toward the appropriate interpretation of alphafold2. *Front. Artif. Intell.*, 6:1149748, August 2023.

Kevin K Yang, Niccolò Zanichelli, and Hugh Yeh. Masked inverse folding with sequence transfer for protein representation learning. *Protein Engineering, Design and Selection*, 36:gzad015, 10 2022. ISSN 1741-0126. doi: 10.1093/protein/gzad015. URL `https://doi.org/10.1093/protein/gzad015`.

Kevin K. Yang, Nicolo Fusi, and Alex X. Lu. Convolutions are competitive with transformers for protein sequence pretraining. *bioRxiv*, 2023a. doi: 10.1101/2022.05.19.492714. URL `https://www.biorxiv.org/content/early/2023/02/23/2022.05.19.492714`.

Wei Yang and Luhua Lai. Computational design of ligand-binding proteins. *Current Opinion in Structural Biology*, 45:67–73, 2017. ISSN 0959-440X. doi: https://doi.org/10.1016/j.sbi.2016.11.021. URL `https://www.sciencedirect.com/science/article/pii/S0959440X16301464`. Engineering and design: New trends in designer proteins.

Zhenyu Yang, Xiaoxi Zeng, Yi Zhao, and Runsheng Chen. AlphaFold2 and its applications in the fields of biology and medicine. *Signal Transduct. Target. Ther.*, 8(1):115, March 2023b.

Yang Zhang and Jeffrey Skolnick. Scoring function for automated assessment of protein structure template quality. *Proteins*, 57(4):702–710, December 2004.

Liangzhen Zheng, Jingrong Fan, and Yuguang Mu. Onionnet: a multiple-layer intermolecular-contact-based convolutional neural network for protein–ligand binding affinity prediction. *ACS Omega*, 4(14):15956–15965, 2019. doi: 10.1021/acsomega.9b01997. URL `https://doi.org/10.1021/acsomega.9b01997`. PMID: 31592466.

Zaixiang Zheng, Yifan Deng, Dongyu Xue, Yi Zhou, Fei Ye, and Quanquan Gu. Structure-informed language models are protein designers. In *Proceedings of the 40th International Conference on Machine Learning*, ICML'23. JMLR.org, 2023.

Cheyenne Ziegler, Jonathan Martin, Claude Sinner, and Faruck Morcos. Latent generative landscapes as maps of functional diversity in protein sequence space. *Nat. Commun.*, 14(1):2222, April 2023.

## A    Benchmarking ProteinReDiff against Related Models

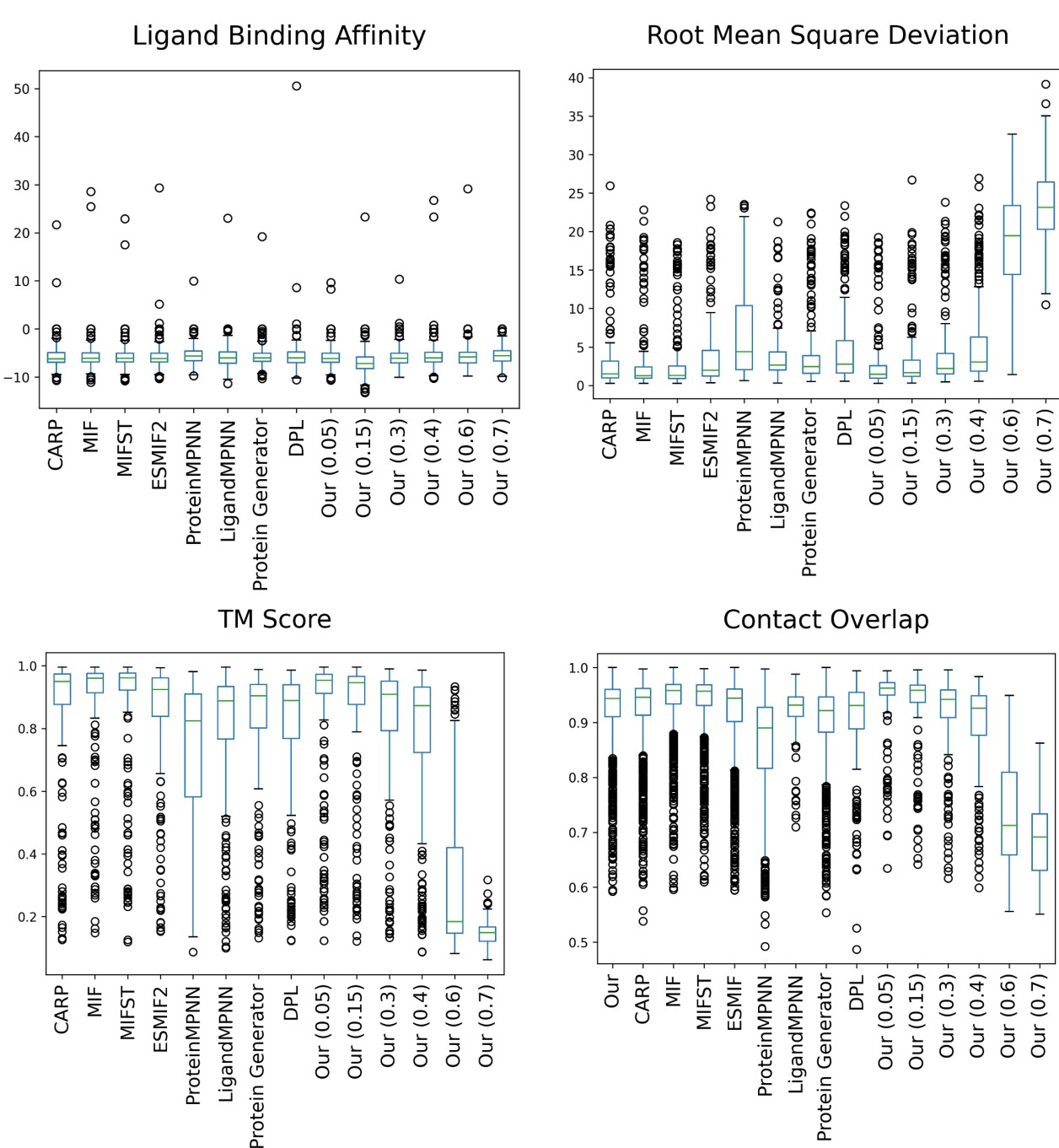

Figure 10: Boxplot illustrating the distribution of ligand binding affinities, and structure preservation metrics (TM Score and RMSD) across all methods evaluated, including baseline models and variations of ProteinReDiff. Each boxplot showcases the median, quartiles, and outliers within the data, providing insight into the variability and central tendency of each metric across the dataset's samples.

## B  Evaluating Protein-Ligand Complex Representation

**Evaluation Methodology**   In the continuation of our study's exploration of protein-ligand complex representations, we extended the use of the PDBBind v2020 dataset, previously detailed in our training process, to evaluate the effectiveness of embeddings generated by ProteinReDiff. Employing these embeddings as input features, we trained a Gaussian Process (GP) model aimed at predicting ligand binding affinity. The choice of a GP model, recognized for its probabilistic nature and adaptability to the nuanced, uncertain dynamics of biological interactions, was pivotal in assessing how well our embeddings encapsulate predictive information about protein-ligand interactions. The GP model used a Gaussian likelihood, which is appropriate for regression tasks, along with a Radial Basis Function (RBF) kernel. We chose the RBF kernel due to its effectiveness in modeling smooth, continuous variations, which is characteristic of protein-ligand binding affinities. The training of the GP model focused on optimizing the parameters to ensure a robust fit to the training data.

Table 5: Experimental results of ligand binding affinity prediction task on PDBBind v2020 dataset.

| Approach | RMSE ↓ ($-\log K_d/K_i$) | MAE ↓ ($-\log K_d/K_i$) | Pearson ↑ | Spearman ↑ |
|---|---|---|---|---|
| Pafnucy (Stepniewska-Dziubinska et al., 2018) | 1.435 | 1.144 | 0.635 | 0.587 |
| OnionNet (Zheng et al., 2019) | 1.403 | 1.103 | 0.648 | 0.602 |
| IGN (Jiang et al., 2021) | 1.404 | 1.116 | 0.662 | 0.638 |
| SIGN (Li et al., 2021) | 1.373 | 1.086 | 0.685 | 0.656 |
| SMINA (Koes et al., 2013) | 1.466 | 1.161 | 0.665 | 0.663 |
| GNINA (McNutt et al., 2021) | 1.740 | 1.413 | 0.495 | 0.494 |
| dMaSIF (Sverrisson et al., 2021) | 1.450 | 1.136 | 0.629 | 0.588 |
| TankBind (Lu et al., 2022) | 1.345 | 1.060 | 0.718 | **0.689** |
| GraphDTA (Nguyen et al., 2020) | 1.564 | 1.223 | 0.612 | 0.570 |
| TransCPI (Chen et al., 2020) | 1.493 | 1.201 | 0.604 | 0.551 |
| MolTrans (Huang et al., 2020) | 1.599 | 1.271 | 0.539 | 0.474 |
| DrugBAN (Bai et al., 2023a) | 1.480 | 1.159 | 0.657 | 0.612 |
| DGraphDTA (Jiang et al., 2020) | 1.493 | 1.201 | 0.604 | 0.551 |
| WGNN-DTA (Bai et al., 2023b) | 1.501 | 1.196 | 0.605 | 0.562 |
| STAMP-DPI (Wang et al., 2022) | 1.503 | 1.176 | 0.653 | 0.601 |
| PSICHIC (Koh et al., 2023) | **1.314** | **1.015** | 0.710 | 0.686 |
| ProteinReDiff (Our) | 1.443 | 1.168 | **0.721** | 0.639 |

## C  Ablation Studies

### C.1  Mask ablations

Table 6: Ablation Study Results on Mask Ratios. The table shows the impact of different mask ratios on validation and test set performance metrics.

| Mask Ratio | Valid | | | | | Test | | | | |
|---|---|---|---|---|---|---|---|---|---|---|
| | LBA ↓ | Sequence Diversity ↑ | TM-Score ↑ | RMSD ↓ | CO ↑ | LBA ↓ | Sequence Diversity ↑ | TM-Score ↑ | RMSD ↓ | CO ↑ |
| 5% | -4.602 ± 0.377 | 87.252 | 0.555 ± 0.023 | 8.225 ± 0.510 | 0.788 ± 0.008 | -6.058 ± 0.182 | 180.800 | 0.734 ± 0.025 | **6.685** ± 0.629 | 0.879 ± 0.010 |
| 10% | -4.410 ± 0.541 | 89.472 | **0.598** ± 0.022 | **7.808** ± 0.544 | **0.873** ± 0.008 | -6.101 ± 0.194 | 184.564 | **0.739** ± 0.027 | 7.108 ± 0.784 | **0.883** ± 0.010 |
| 15% | **-4.890** ± 0.303 | 89.601 | 0.581 ± 0.022 | 8.252 ± 0.537 | 0.867 ± 0.008 | **-6.202** ± 0.167 | 184.925 | 0.729 ± 0.025 | 7.257 ± 0.768 | 0.877 ± 0.010 |
| 30% | -4.596 ± 0.257 | **90.643** | 0.453 ± 0.022 | 10.707 ± 0.604 | 0.820 ± 0.008 | -5.553 ± 0.188 | 181.978 | 0.221 ± 0.015 | 21.166 ± 0.740 | 0.707 ± 0.009 |
| 40% | -4.668 ± 0.281 | 89.091 | 0.297 ± 0.016 | 14.309 ± 0.497 | 0.768 ± 0.008 | -5.794 ± 0.286 | 185.136 | 0.390 ± 0.024 | 15.014 ± 0.717 | 0.750 ± 0.011 |
| 50% | -4.052 ± 1.162 | 90.445 | 0.390 ± 0.020 | 10.886 ± 0.424 | 0.788 ± 0.009 | -6.034 ± 0.177 | **188.163** | 0.567 ± 0.029 | 10.239 ± 0.688 | 0.807 ± 0.012 |
| 60% | -4.678 ± 0.262 | 88.643 | 0.226 ± 0.011 | 14.142 ± 0.337 | 0.729 ± 0.007 | -5.981 ± 0.258 | 184.356 | 0.243 ± 0.017 | 18.092 ± 0.525 | 0.702 ± 0.009 |
| 70% | -4.214 ± 0.264 | 81.333 | 0.165 ± 0.004 | 18.226 ± 0.456 | 0.733 ± 0.007 | -5.360 ± 0.175 | 162.841 | 0.145 ± 0.004 | 24.944 ± 0.646 | 0.689 ± 0.008 |

### C.2  Featurization and model architecture ablations

Table 7: Ablation Setup of Featurization and Model Architecture

|  |  | No bond distance | No bond feats | No bond | No ligand | No ligand, only ESM | No ESM | No SRA | No OPU | DPL (No SRA/OPU) |
|---|---|---|---|---|---|---|---|---|---|---|
|  |  | Ablation studies | | | | | | | | |
| Ligand | Bond distance |  | ✓ |  |  |  | ✓ | ✓ | ✓ | ✓ |
|  | Bond feats (type, ring, etc.) | ✓ |  |  |  |  | ✓ | ✓ | ✓ | ✓ |
|  | Ligand atom feats (chirality, charge, degree, etc.) | ✓ | ✓ | ✓ |  |  | ✓ | ✓ | ✓ | ✓ |
| Protein | ESM embeddings | ✓ | ✓ | ✓ | ✓ | ✓ |  | ✓ | ✓ | ✓ |
|  | Residue feats (pos. encodings, res. type) | ✓ | ✓ | ✓ | ✓ |  | ✓ | ✓ | ✓ | ✓ |
| Model architecture | Single Representation Attention (SRA) | ✓ | ✓ | ✓ | ✓ | ✓ | ✓ |  | ✓ |  |
|  | Outer Product Update (OPU) | ✓ | ✓ | ✓ | ✓ | ✓ | ✓ | ✓ |  |  |

Table 8: Ablation Study Results on Input Featurization Methods. The table presents the impact of various feature removals on performance metrics.

| Features | LBA ↓ | Sequence Diversity ↑ | TM-Score ↑ | RMSD ↓ | CO ↑ |
|---|---|---|---|---|---|
| Reference | **-4.890** $\pm$ 0.303 | 89.601 | **0.581** $\pm$ 0.022 | **8.252** $\pm$ 0.537 | **0.877** $\pm$ 0.008 |
| No bond | -4.549 $\pm$ 0.272 | 84.837 | 0.287 $\pm$ 0.016 | 14.325 $\pm$ 0.491 | 0.761 $\pm$ 0.009 |
| No bond distance | -4.869 $\pm$ 0.277 | **90.186** | 0.447 $\pm$ 0.021 | 11.068 $\pm$ 0.579 | 0.821 $\pm$ 0.008 |
| No bond feats | -4.985 $\pm$ 0.289 | 85.974 | 0.475 $\pm$ 0.022 | 9.748 $\pm$ 0.476 | 0.811 $\pm$ 0.010 |
| No ESM | -2.723 $\pm$ 0.176 | 32.222 | 0.136 $\pm$ 0.007 | 37.322 $\pm$ 1.032 | 0.748 $\pm$ 0.008 |
| No ligand | -4.478 $\pm$ 0.252 | 87.723 | 0.324 $\pm$ 0.018 | 14.125 $\pm$ 0.571 | 0.780 $\pm$ 0.008 |
| No OPU | -3.197 $\pm$ 0.304 | 71.669 | 0.102 $\pm$ 0.006 | 40.969 $\pm$ 1.246 | 0.723 $\pm$ 0.004 |
| No SRA | -4.878 $\pm$ 0.282 | 87.054 | 0.424 $\pm$ 0.023 | 11.391 $\pm$ 0.556 | 0.810 $\pm$ 0.009 |
| DPL | -4.153 $\pm$ 0.631 | 86.379 | 0.311 $\pm$ 0.019 | 13.931 $\pm$ 0.527 | 0.744 $\pm$ 0.009 |
| No ligand, only ESM | -4.429 $\pm$ 0.270 | 88.481 | 0.390 $\pm$ 0.020 | 13.108 $\pm$ 0.702 | 0.813 $\pm$ 0.007 |

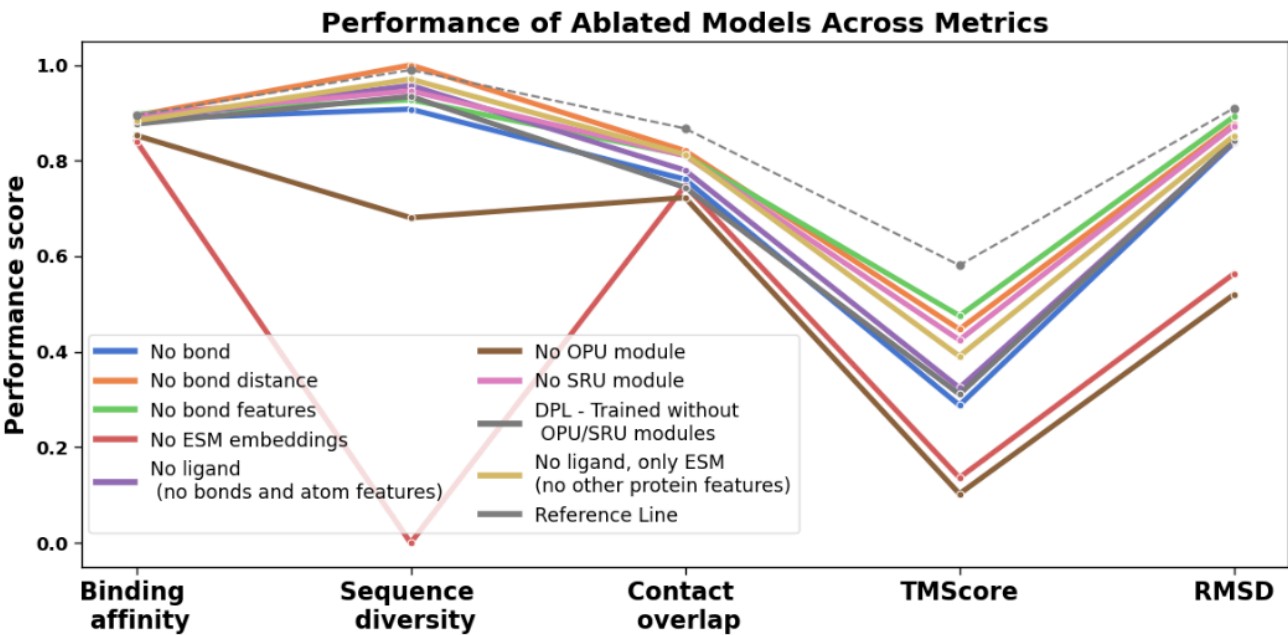

Figure 11: Breakdown of metrics for ablation models based on different featurization methods and architectural adjustments. The dashed line indicates the baseline ProteinReDiff model trained without any ablations.

