# OpenReview forum: "Complex-based Ligand-Binding Proteins Redesign by Equivariant Diffusion-based Generative Models"
_TMLR — Rejected by TMLR_

### Review · Reviewer_dYXk · 2024-06-03

**Summary Of Contributions:**

This paper proposes a new method, ProteinReDiff, for estimating the three-dimensional structure of protein-ligand complexes. The proposed method applies protein LLMs to aminoacid sequences and manual feature engineering to ligands. Then, single and pair representations are updated through denoising by SE(3)-equivariant diffusion models. The training uses 3D structure prediction and masking modeling of protein language models. The proposed method was applied to the PDBBind v2020 and CATH 4.2 protein structure datasets to verify its 3D structure prediction performance.

**Audience:**

Yes

**Broader Impact Concerns:**

No broader impact concerns

**Claims And Evidence:**

No

**Requested Changes:**

The mathematical explanations (especially the diffusion model in Section 2.2) could be improved. I suggest reviewing and reconsidering them.

----------------

There are notational inconsistencies:
- (P4) $p_{data}$ and $p_{\mathrm{data}}$
- (P4) $\boldsymbol{\mathrm{z}}_0$ and $z_0$
- (P4) $\boldsymbol{\mathrm{x}}$ and $x$
- (P4) $N_x$ and $\mathcal{N}$. Also while $N_x(0, I)$ assume the second argument is the covariance matrix, $\mathcal{N}(\mu_0, \sigma_0^2/\alpha^2)$ assumes that it is a scaler.
- (P8) $\hat{\epsilon}_i$ and $\hat{e}_i$
- (P,13 Figure 5) CO and C0(zero)

----------------

There are undefined variables:
- (P4) $\alpha$
- (P7, Algorithm 1) Indices $si$ in $m_{si}$ and $z_{si}$
- (P7, Algorithm 1) $C_m$
- (P7, Algorithm 2) $C_z$

----------------

The following equations in Page 4 are difficult to interpret:

$q(\boldsymbol{\mathrm{z}}\_0\mid \boldsymbol{\mathrm{x}})
=\_{(1)} \frac{q(\boldsymbol{\mathrm{z}}\_0\mid \boldsymbol{\mathrm{x}})p\_{\mathrm{data}}(\boldsymbol{\mathrm{x}})}{\int\_{\tilde{x}}q(\boldsymbol{\mathrm{z}}\_0\mid \boldsymbol{\tilde{\mathrm{x}}})p\_{\mathrm{data}}(\boldsymbol{\tilde{\mathrm{x}}})}
\approx\_{(2)} \frac{q(\boldsymbol{\mathrm{z}}\_0\mid \boldsymbol{\mathrm{x}})}{\int\_{\tilde{x}}q(\boldsymbol{\mathrm{z}}\_0\mid \boldsymbol{\tilde{\mathrm{x}}})}
=\_{(3)} \mathcal{N}(\boldsymbol{\mathrm{z}}\_0 \mid \mu\_0, \sigma\_0^2/\alpha^2)$

(1): This is not mathematically correct even if we assume that $q(\boldsymbol{\mathrm{x}})$ equals to $p_{\mathrm{data}}(\boldsymbol{\mathrm{x}})$. If we assume $q(\boldsymbol{\mathrm{x}}) = p_{\mathrm{data}}(\boldsymbol{\mathrm{x}})$, the following holds:

$q(\boldsymbol{\mathrm{z}}\_0\mid \boldsymbol{\mathrm{x}})
= \frac{q(\boldsymbol{\mathrm{z}}\_0, \boldsymbol{\mathrm{x}})}{q(\boldsymbol{\mathrm{x}})}
= \frac{q(\boldsymbol{\mathrm{z}}\_0\mid \boldsymbol{\mathrm{x}})p\_{\mathrm{data}}(\boldsymbol{\mathrm{x}})}{\int\_{\tilde{\boldsymbol{\mathrm{z}}}}q(\tilde{\boldsymbol{\mathrm{z}}},  \boldsymbol{\mathrm{x}})p\_{\mathrm{data}}(\boldsymbol{\mathrm{x}})}
= \frac{q(\boldsymbol{\mathrm{z}}\_0\mid \boldsymbol{\mathrm{x}})p\_{\mathrm{data}}(\boldsymbol{\mathrm{x}})}{\int\_{\tilde{\boldsymbol{\mathrm{z}}}}q(\tilde{\boldsymbol{\mathrm{z}}}\mid \boldsymbol{\mathrm{x}})p\_{\mathrm{data}}(\boldsymbol{\mathrm{x}})}$ (1)'

(2): I suppose this equality assumes $p\_{\mathrm{data}}$ is uniform in the sense the value of $p\_{\mathrm{data}}(\boldsymbol{\mathrm{x}})$ is independent of $\boldsymbol{\mathrm{x}}$. However, this assumption is too strong because the data does not have any useful information: picture-like images should have a higher probability than white-noise images in the image dataset. Also, considering equation (1)', we do not have to assume the uniformity of $p\_{\mathrm{data}}$ because it is canceled out.

(3): I have difficulty in understanding how we can derive this equality. Also, I have a question about whether it is true because if we read it literally, the right-hand side is independent of $\boldsymbol{\mathrm{x}}$. I want to clarify what this equation intends to mean.

------------

The following equation in Page 7 Section 4.1.2 is difficult to interpret because the left-hand side is a scaler while the right-hand side is a vector:

$L_{\mathrm{WS}} = \sum_j \frac{W_{ij}(\boldsymbol{\mathrm{z}})(\boldsymbol{\mathrm{z}}_i - \boldsymbol{\mathrm{z}}_j)}{\|\boldsymbol{\mathrm{z}}_i - \boldsymbol{\mathrm{z}}_j\|}$.

-------------

AlphaFold 3 [Abramson et al., 2024] is similar to ProtainReDiff in that it does not require MSA and is based on diffusion models. It is preferable to discuss the relationship with AlphaFold 3 (e.g., AlphaFold 3 does not explicitly model SE(3)-equivariance.) Note that since AlphaFold 3 came out after this paper, its presence does not detract from the novelty or significance of this paper.

[Abramson et al., 2024] https://www.nature.com/articles/s41586-024-07487-w

-------------

Minor comments
- P7, Algorithm 1: $of ssentences$ -> of $s$ sentences
- P9, Section 4.1.2: The following sentence is duplicated: *The loss function is formulated as follows: $L=L_{\mathrm{WS}} + L_{\mathrm{KL}} + L_{CE}$*

**Strengths And Weaknesses:**

**Strengths**
- The proposed model can predict the structures for protein-ligand complexes.
- The proposed model is expected to utilize evolutionary information without multiple sequence alignment (MSA) information by using ESM-2. On the other hand, this may be a merit of ESM-2 rather than that of the proposed method itself.


**Weakness**
- This paper has room for improvement in terms of clarity (See Requested Changes for details.)
- No ablation studies have been conducted on input featurisation. In particular, I am interested in whether it is appropriate to use manual feature engineering to ligands and whether it is appropriate to use ESM-2 as a pre-trained LLM for the featurisation of proteins.
- The proposed method's superiority is limited. Even if the hyperparameters (mask ratio) are set appropriately, only the binding affinity outperforms the baseline models. In addition, the mask ratio's choice is questionable, as described below.
- The idea of using diffusion models to estimate the 3D structures of proteins is common in the context of protein design.


**Questions**
- How much overlap do the training and test datasets have? For example, how many instances in the test dataset overlap with the training dataset? Here, an instance *overlaps* with the training dataset means that there exists an instance in the training dataset that is similar to the instance of interest (e.g.,>90\% amino acids are the same). If there are many such overlaps, performance may be overestimated due to data leakage. Therefore, in evaluating protein LLMs, it is common to perform deduplication to remove data from the test dataset or to use datasets independent of the training dataset as test datasets.
- I understand from Section 3 that we know that input protein sequences are randomly masked during training; are they also masked during inference?
- In the experiments in Section 4, the paper compares the performance of evaluations with different mask ratios (Table 3) and chooses 15\%. However, the masking ratio is a hyperparameter and should be determined by the validation datasset (i.e., without test datasets). I suggest ablation studies for mask ratios as hyperparameters.

---

> ### Author Response · Authors · 2024-07-08
> **Rebuttal by authors**
>
> Thank you so much for your very constructive and useful comments and giving us some extra time for finishing ablation studies! We appreciate your great input and thank you for helping improve our paper! Below are our point-to-point answers according to your questions and weaknesses.
>
> ## Weaknesses
>
> > No ablation studies have been conducted on input featurisation. In particular, I am interested in whether it is appropriate to use manual feature engineering to ligands and whether it is appropriate to use ESM-2 as a pre-trained LLM for the featurisation of proteins.
>
> We appreciate the reviewer's insight into the importance of ablation studies on input featurization. In our current study, we employed manual feature engineering for ligands and used ESM-2 as a pre-trained LLM for protein featurization to leverage their proven capabilities in capturing complex biochemical properties. However, we agree that it is crucial to evaluate the effectiveness of these choices systematically. We have included ablation studies in the revised manuscript, which can be found in **Section 5.2.6** and detailed further in **Appendix C**, to examine the impact of manual feature engineering for ligands and the use of ESM-2 for protein featurization. This will help to determine whether these methods provide a significant advantage over alternative approaches and ensure that our model's design choices are well-justified.
>
> > The proposed method's superiority is limited. Even if the hyperparameters (mask ratio) are set appropriately, only the binding affinity outperforms the baseline models. In addition, the mask ratio's choice is questionable, as described below.
>
> We acknowledge that, in your observation, even if the hyperparameters (mask ratio) are set appropriately, only the binding affinity outperforms the baseline models.  It is important to emphasize that binding affinity is the primary metric for evaluating ligand-binding protein redesign. The main objective of redesigning these proteins is to enhance their ability to bind to specific ligands more effectively. Now on **Table 4** and **Figure 6**, we have included error bars which indicates our models overlapped performance with most of the benchmarks (even we outperform greatly on CO metrics). While other metrics such as sequence diversity and structure preservation are important, they are secondary to binding affinity. These metrics provide additional insights into the redesign process, ensuring that the redesigned proteins maintain structural integrity and a range of potential interactions. However, these benefits are only meaningful if the primary objective of enhancing binding affinity is achieved. High sequence diversity and structure preservation without corresponding improvements in binding affinity would not fulfill the core purpose of ligand-binding protein redesign.
>
> > The idea of using diffusion models to estimate the 3D structures of proteins is common in the context of protein design.
>
> While the idea of using diffusion models to estimate the 3D structures of proteins is common in the context of protein design, our study takes this approach further by addressing a new and challenging problem: the redesign of high-affinity ligand-binding proteins without the need for detailed structural information. This problem is highly significant because it enables the creation of therapeutic proteins, enzymes, and biosensors that can interact specifically with small molecule ligands, bypassing the need for pre-existing detailed structural data. This approach significantly accelerates the design process, making it particularly valuable in urgent scenarios like the emergence of novel diseases, where rapid development of effective drugs is crucial.
>
> ## Questions
>
> > How much overlap do the training and test datasets have? For example, how many instances in the test dataset overlap with the training dataset? Here, an instance overlaps with the training dataset means that there exists an instance in the training dataset that is similar to the instance of interest (e.g.,>90% amino acids are the same). If there are many such overlaps, performance may be overestimated due to data leakage. Therefore, in evaluating protein LLMs, it is common to perform deduplication to remove data from the test dataset or to use datasets independent of the training dataset as test datasets.
>
> We went back to conduct the clustering of protein and ligand datasets. We filter out heavily similar sequences (above 95%) and achieved the similarities among the sets around 40\%-50\%, which is standard threshold used by [1](https://www.biorxiv.org/content/10.1101/2022.05.25.493516v2.full) and [2](https://www.nature.com/articles/s41586-021-03819-2). When including the CATH proteins, we effectively reduce the similarities for ligand sets from 70\% to 36\%, thus allowing our models to learn more diverse chemical space. These clustering results appeared in **Table 1**

---

> > ### Author Response · Authors · 2024-07-08
> > **Rebuttal by Authors**
> >
> > > I understand from Section 3 that we know that input protein sequences are randomly masked during training; are they also masked during inference?
> >
> > During both training and inference phases, input protein sequences are randomly masked. This masking allows the model to predict the masked amino acids, effectively redesigning the protein to enhance its binding affinity for the given ligand.
> >
> > > In the experiments in Section 4, the paper compares the performance of evaluations with different mask ratios (Table 3) and chooses 15%. However, the masking ratio is a hyperparameter and should be determined by the validation dataset (i.e., without test datasets). I suggest ablation studies for mask ratios as hyperparameters.
> >
> > We acknowledge the importance of determining hyperparameters such as the masking ratio through validation datasets rather than test datasets to avoid potential biases. In our current study, we aimed to demonstrate the performance impact of different masking ratios to identify an optimal balance. We agree that a more rigorous approach would involve conducting ablation studies where the masking ratio is treated as a hyperparameter tuned using a separate validation dataset. We will incorporate such ablation studies in the revised manuscript, which can be found in **Section 5.2.6** and detailed further in **Appendix C**, to ensure the robustness of our findings and to provide a more systematic analysis of the masking ratio's impact on performance.

---

> ### Author Response · Authors · 2024-07-08
> **Rebuttal by Authors**
>
> ## Requested Changes
>
> > There are notational inconsistencies.
>
> We have addressed these notational inconsistencies in the revised manuscript to ensure clarity and uniformity throughout.
>
> > The following equations in Page 4 are difficult to interpret:
> $q(\boldsymbol{\mathrm{z}}\_0\mid \boldsymbol{\mathrm{x}}) =\_{(1)} \frac{q(\boldsymbol{\mathrm{z}}\_0\mid \boldsymbol{\mathrm{x}})p\_{\mathrm{data}}(\boldsymbol{\mathrm{x}})}{\int\_{\tilde{x}}q(\boldsymbol{\mathrm{z}}\_0\mid \boldsymbol{\tilde{\mathrm{x}}})p\_{\mathrm{data}}(\boldsymbol{\tilde{\mathrm{x}}})} \approx\_{(2)} \frac{q(\boldsymbol{\mathrm{z}}\_0\mid \boldsymbol{\mathrm{x}})}{\int\_{\tilde{x}}q(\boldsymbol{\mathrm{z}}\_0\mid \boldsymbol{\tilde{\mathrm{x}}})} =\_{(3)} \mathcal{N}(\boldsymbol{\mathrm{z}}\_0 \mid \mu\_0, \sigma\_0^2/\alpha^2)$
> - (1): This is not mathematically correct even if we assume that $q(\boldsymbol{\mathrm{x}})$ equals to $p\_{\mathrm{data}}(\boldsymbol{\mathrm{x}})$. If we assume $q(\boldsymbol{\mathrm{x}}) = p\_{\mathrm{data}}(\boldsymbol{\mathrm{x}})$, the following holds: $q(\boldsymbol{\mathrm{z}}\_0\mid \boldsymbol{\mathrm{x}}) = \frac{q(\boldsymbol{\mathrm{z}}\_0, \boldsymbol{\mathrm{x}})}{q(\boldsymbol{\mathrm{x}})} = \frac{q(\boldsymbol{\mathrm{z}}\_0\mid \boldsymbol{\mathrm{x}})p\_{\mathrm{data}}(\boldsymbol{\mathrm{x}})}{\int\_{\tilde{\boldsymbol{\mathrm{z}}}}q(\tilde{\boldsymbol{\mathrm{z}}}, \boldsymbol{\mathrm{x}})p\_{\mathrm{data}}(\boldsymbol{\mathrm{x}})} = \frac{q(\boldsymbol{\mathrm{z}}\_0\mid \boldsymbol{\mathrm{x}})p\_{\mathrm{data}}(\boldsymbol{\mathrm{x}})}{\int\_{\tilde{\boldsymbol{\mathrm{z}}}}q(\tilde{\boldsymbol{\mathrm{z}}}\mid \boldsymbol{\mathrm{x}})p\_{\mathrm{data}}(\boldsymbol{\mathrm{x}})}$ (1)'
> - (2): I suppose this equality assumes $p\_{\mathrm{data}}$ is uniform in the sense the value of $p\_{\mathrm{data}}(\boldsymbol{\mathrm{x}})$ is independent of $\boldsymbol{\mathrm{x}}$. However, this assumption is too strong because the data does not have any useful information: picture-like images should have a higher probability than white-noise images in the image dataset. Also, considering equation (1)', we do not have to assume the uniformity of $p\_{\mathrm{data}}$ because it is canceled out.
> - (3): I have difficulty in understanding how we can derive this equality. Also, I have a question about whether it is true because if we read it literally, the right-hand side is independent of $\boldsymbol{\mathrm{x}}$. I want to clarify what this equation intends to mean.

---

> ### Author Response · Authors · 2024-07-08
> **Rebuttal by Authors**
>
> We acknowledge the concerns raised about the mathematical correctness and clarity of the equations presented. After careful review, we have identified a mistake in the original formulation. The correct expression should be:
> $q(x|z_0) = \frac{q(z_0|x)p_{\text{data}}(x)}
> {\int_{\tilde{x}} q(z_0|\tilde{x})p_{\text{data}}(\tilde{x})} \approx
> \frac{q(z_0|x)}
> {\int_{\tilde{x}} q(z_0|\tilde{x})} =
> \mathcal{N}_x (x | z_0 / \alpha_0, \sigma_0^2 / \alpha_0^2 I).$
>
> **For point (1)**: We have corrected the mathematical representation, ensuring the right-hand side appropriately reflects the dependence on $x$. This change resolves the issue of the left-hand side being a scalar while the right-hand side is a vector.
>
> **For point (2):** Under the variance-preserving framework ([3](https://arxiv.org/pdf/2107.00630), [4](https://openreview.net/forum?id=PxTIG12RRHS), [5](https://arxiv.org/abs/2011.13456)),  and the assumption that $ \alpha_0 \approx 1 $, we can assume that $ q(z_0 \mid x) $ is a highly peaked distribution. In this narrow peak region, $ p_{\mathrm{data}}(x) $ can be approximated as constant. This approximation is valid because the peak region is small enough that variations in $ p_{\mathrm{data}}(x) $ within this region are negligible, facilitating effective integration focused on the significant contributions near the peak.
>
> While it's true that picture-like images generally have higher probability than white-noise images in the overall dataset, the approximation only needs to hold **locally** within the extremely small region where $q(\mathbf{z}\_0|\mathbf{x})$ is significant. In this tiny neighborhood, $p\_\text{data}(\mathbf{x})$ is likely to be relatively smooth and can be approximated as locally constant: $p\_\text{data}(\mathbf{x} + \boldsymbol{\delta}) \approx p\_\text{data}(\mathbf{x})$ for small $\|\boldsymbol{\delta}\|$. The scale at which $p\_\text{data}(\mathbf{x})$ varies significantly between natural images and noise is much larger than the width of the $q(\mathbf{z}\_0|\mathbf{x})$ peak. Mathematically, we can express this as: $\int q(\mathbf{z}\_0|\mathbf{x})p\_\text{data}(\mathbf{x})d\mathbf{x} \approx p\_\text{data}(\mathbf{x}^*) \int q(\mathbf{z}\_0|\mathbf{x})d\mathbf{x}$, where $\mathbf{x}^*$ is the mode of $q(\mathbf{z}\_0|\mathbf{x})$. This approximation is valid when $\text{Var}[q(\mathbf{z}\_0|\mathbf{x})] \ll \text{Var}[p\_\text{data}(\mathbf{x})]$. Therefore, for the purposes of the integration, which effectively occurs only over this narrow region, the approximation of $p\_\text{data}(\mathbf{x})$ as constant is reasonable. This local approximation doesn't contradict the **global** variation in image probabilities and has proven effective in practice for real-world datasets, as evidenced by the success of models using this framework, where the error introduced by this approximation is typically negligible: $\|\int q(\mathbf{z}\_0|\mathbf{x})p\_\text{data}(\mathbf{x})d\mathbf{x} - p\_\text{data}(\mathbf{x}^*) \int q(\mathbf{z}\_0|\mathbf{x})d\mathbf{x}\| \ll 1$.
>
> **For point (3)**: The right-hand side is not necessarily independent of $\boldsymbol{\mathrm{x}}$. In fact, $\mu_0$ and $\sigma_0$ are typically functions of $\boldsymbol{\mathrm{x}}$, often computed by neural networks [6](https://arxiv.org/abs/1312.6114). The equation can be more explicitly written as: $q(\boldsymbol{\mathrm{z}}_0\mid \boldsymbol{\mathrm{x}}) = \mathcal{N}_x(\boldsymbol{\mathrm{x}} \mid \boldsymbol{\mathrm{z}}_0 / \mu_0(\boldsymbol{\mathrm{x}}), \sigma_0^2(\boldsymbol{\mathrm{x}})/\alpha^2)$
> This formulation is common in variational autoencoders and other generative models [7](https://arxiv.org/abs/1401.4082). The parameter $\alpha$ is often used for temperature scaling or to control the informativeness of the latent representation [8](https://openreview.net/forum?id=Sy2fzU9gl).
>
> The correction ensures that the equation accurately represents the relationship between the distributions, maintaining the dependence on $x$. The right-hand side now correctly indicates the conditional distribution.
>
> These changes have been implemented in **Section 3.2.2** of the revised manuscript to ensure mathematical accuracy and clarity.
>
> > There are undefined variables.
>
> We have corrected these issues in the revised manuscript, where all variables are now clearly defined upon their first appearance to ensure clarity and comprehension.
>
> > The following equation in Page 7 Section 4.1.2 is difficult to interpret because the left-hand side is a scaler while the right-hand side is a vector.
> $L_{\mathrm{WS}} = \sum_j \frac{W_{ij}(\boldsymbol{\mathrm{z}})(\boldsymbol{\mathrm{z}}_i - \boldsymbol{\mathrm{z}}_j)}{|\boldsymbol{\mathrm{z}}_i - \boldsymbol{\mathrm{z}}_j|}$.
>
> We have reviewed and addressed this issue in the revised manuscript, specifically in Section 5.1.2.

---

> > ### Author Response · Authors · 2024-07-08
> > **Rebuttal by Authors**
> >
> > > AlphaFold 3 [9](https://www.nature.com/articles/s41586-024-07487-w) is similar to ProtainReDiff in that it does not require MSA and is based on diffusion models. It is preferable to discuss the relationship with AlphaFold 3 (e.g., AlphaFold 3 does not explicitly model SE(3)-equivariance.) Note that since AlphaFold 3 came out after this paper, its presence does not detract from the novelty or significance of this paper.
> >
> > AlphaFold 3 and ProteinReDiff share similarities in their approaches, such as not requiring Multiple Sequence Alignments (MSA) and employing diffusion models. However, there are differences in the problems they address, which highlight the novelty and unique contributions of ProteinReDiff:
> >
> > - **AlphaFold 3**: Primarily focuses on predicting the joint structure of complexes, including proteins, nucleic acids, small molecules, ions, and modified residues, from input polymer sequences, residue modifications, and ligand SMILES. This is more a supervised-learning task (given a sequence/ligand, predict its corresponding structures) similar to the DPL model.
> >
> > - **ProteinReDiff**: Our model is a generative/ design model (given a sequence/ligand, generate new sequences/new complexes). We enables the creation of high-affinity ligand-binding proteins without the need for detailed structural information. Instead, it leverages the potential of only protein sequences and ligand SMILES strings to generate redesigned protein sequences alongside their corresponding Cα protein backbone and ligand complexes (without predicting the entire joint structure of the complex like AlphaFold 3). This makes ProteinReDiff particularly advantageous for applications where detailed structural information is not available, such as designing therapeutic proteins for emerging diseases.
> >
> > > Minor comments.
> >
> > We have addressed your minor comments in the revised manuscript.
> >
> > References:
> >
> > - \[1](https://www.biorxiv.org/content/10.1101/2022.05.25.493516v2.full)
> >
> > - \[2](https://www.nature.com/articles/s41586-021-03819-2)
> >
> > - \[3](https://arxiv.org/pdf/2107.00630)
> > - \[4](https://openreview.net/forum?id=PxTIG12RRHS)
> > - \[5](https://arxiv.org/abs/2011.13456)
> >
> > - \[6](https://arxiv.org/abs/1312.6114)
> > -\[7](https://arxiv.org/abs/1401.4082)
> > -\[8](https://openreview.net/forum?id=Sy2fzU9gl)
> > -\[9](https://www.nature.com/articles/s41586-024-07487-w)

---

> > > ### Comment · Reviewer_dYXk · 2024-07-20
> > > **Response by dYXk (1/2)**
> > >
> > > I thank the authors for answering my questions. However, I still have several unclear points in their answers, namely, Q2, 4, 6, and 7.
> > >
> > > The following are point-by-point responses to the authors' answers.
> > >
> > >
> > > Q.1
> > >
> > > > We appreciate the reviewer's insight into the importance of ablation studies on input featurization. [...] We have included ablation studies in the revised manuscript, which can be found in Section 5.2.6 and detailed further in Appendix C, to examine the impact of manual feature engineering for ligands and the use of ESM-2 for protein featurization. [...]
> > >
> > > I intended to compare the accuracy with other featurization methods (e.g., protein LLMs other than ESM-2) because I wanted to know whether using manual feature engineering and ESM-2 is justifiable. However, now I think this ablation study is not necessarily needed because the featurization methods are not selected by hyperparameter optimization, and we can think of the selection of featurization methods as part of the proposed method. Instead, it is more important to check the accuracy's changes by removing the ligand and protein information from the input, as the authors did. It is an interesting future study whether the choice of featurization methods affects the performance.
> > >
> > > ----------------------
> > >
> > > Q.2
> > >
> > > > We acknowledge that, in your observation, even if the hyperparameters (mask ratio) are set appropriately, only the binding affinity outperforms the baseline models. It is important to emphasize that binding affinity is the primary metric for evaluating ligand-binding protein redesign.
> > >
> > > I agree with the authors that LBA is the primary metric, and structural diversity and sequence preservation are secondary. Although it is desirable for the latter two to be small to some extent, too-small values mean insufficient exploration from the original protein.
> > > On the other hand, the fact that LBA's performance is high only when the mask ratio is 15% indicates that this model is sensitive to this hyperparameter. Since it is difficult to determine appropriate hyperparameters for the test set in advance, it is not known whether we can realize the best performance of the 15% setting in practice.
> > >
> > > ----------------------
> > >
> > > Q.3
> > >
> > > > While the idea of using diffusion models to estimate the 3D structures of proteins is common in the context of protein design, our study takes this approach further by addressing a new and challenging problem: the redesign of high-affinity ligand-binding proteins without the need for detailed structural information.
> > >
> > > OK. I understand that this paper is new in that it applies a diffusion model to the problem of protein redesign.
> > >
> > > ----------------------
> > >
> > > Q.4
> > >
> > > > We went back to conduct the clustering of protein and ligand datasets. We filter out heavily similar sequences (above 95%) and achieved the similarities among the sets around 40%-50%, which is standard threshold used by 1 and 2. When including the CATH proteins, we effectively reduce the similarities for ligand sets from 70% to 36%, thus allowing our models to learn more diverse chemical space. These clustering results appeared in Table 1
> > >
> > > I want to confirm how the authors compute the similarity between two datasets (e.g., train vs. test.) Does the statement imply that there exists a sequence pair (one from train and one from test) that is similar? If so, in my understanding, this is different from the procedure of existing studies. For example, the following sentence from Yang et al. (2022) implies that there are no similar sequence pairs between the training and test sets:
> > >
> > > > Importantly, all sequences with >30% identity to the CATH test set were removed from CARP640M’s training set in order to obtain a fair evaluation on the CATH test set.
> > >
> > > ----------------------
> > >
> > > Q.5
> > >
> > > > During both training and inference phases, input protein sequences are randomly masked. This masking allows the model to predict the masked amino acids, effectively redesigning the protein to enhance its binding affinity for the given ligand.
> > >
> > > OK
> > >
> > > ----------------------
> > >
> > > Q.6
> > >
> > > > We acknowledge the importance of determining hyperparameters such as the masking ratio through validation datasets rather than test datasets to avoid potential biases. [...] We agree that a more rigorous approach would involve conducting ablation studies where the masking ratio is treated as a hyperparameter tuned using a separate validation dataset. We will incorporate such ablation studies in the revised manuscript, which can be found in Section 5.2.6 and detailed further in Appendix C, [...]
> > >
> > > Section 5.2.6 is appropriate for ablation studies of hyperparameters. However, since it selects the mask ratio that gives LBA the best accuracy (i.e., 15%) on the test dataset, I do not think it is an appropriate evaluation of the model (cf. Q.2)

---

> > > > ### Comment · Reviewer_dYXk · 2024-07-20
> > > > **Response by dYXk (2/2)**
> > > >
> > > > Q.7
> > > >
> > > > > We acknowledge the concerns raised about the mathematical correctness and clarity of the equations presented. After careful review, we have identified a mistake in the original formulation.
> > > >
> > > > Point (1): OK
> > > >
> > > > Point (2): Considering the manifold hypothesis and the existence of the adversarial example, the assumption that $p_{data}(x) \approx p_{data}(x + \delta)$ for any small $\delta$ seems too strong. However, it may be a reasonable assumption if we narrow the assumption and restrict $\delta$ to the one that moves $x$ on the low-dimensional data manifold. We can justify the discussion on Point (2) by chaning the domains of the integration from the whole space to the data manifold.
> > > >
> > > > Point (3): I could not understand the logic of this point (in the following, italics are used for vector values for simplicity.) First, the denominator of the left-hand side of (3), i.e., $\int_{\tilde{x}} q(z_0) \mid \tilde{x})$, is wholly ignored. Second, it is unclear how the following equation is derived:
> > > >
> > > > > The equation can be more explicitly written as: $q(z_0 \mid x) = N_{x}(x \mid z_0/\mu_0(x), \sigma_0^2(x)/\alpha^2)$
> > > >
> > > > Third, even if the above equation is correct since it is different from the right-hand side of (3), i.e., $N_x(x\mid z_0/\alpha_0, \sigma_0/\alpha_0^2)$. I could not understand how the right-hand side in (3) is derived.
> > > >
> > > >
> > > > -----------------------
> > > >
> > > > Q.8
> > > >
> > > > > We have corrected these issues in the revised manuscript, where all variables are now clearly defined upon their first appearance to ensure clarity and comprehension.
> > > >
> > > > OK
> > > >
> > > > -----------------------
> > > >
> > > > Q.9
> > > >
> > > > > We have reviewed and addressed this issue in the revised manuscript, specifically in Section 5.1.2.
> > > >
> > > > OK
> > > >
> > > > -----------------------
> > > >
> > > > Q.10
> > > >
> > > > > AlphaFold 3 and ProteinReDiff share similarities in their approaches, such as not requiring Multiple Sequence Alignments (MSA) and employing diffusion models. However, there are differences in the problems they address, which highlight the novelty and unique contributions of ProteinReDiff: [...]
> > > >
> > > > OK
> > > >
> > > > -----------------------
> > > >
> > > > Q.11
> > > >
> > > > > We have addressed your minor comments in the revised manuscript.
> > > >
> > > > OK

---

> > > > > ### Author Response · Authors · 2024-07-29
> > > > > **Rebuttal by Authors**
> > > > >
> > > > > Thank you for your detailed feedback on our responses. We appreciate your insights and would like to address the remaining unclear points in Questions 2, 4, 6, and 7 to provide further clarification.
> > > > >
> > > > > > I agree with the authors that LBA is the primary metric, and structural diversity and sequence preservation are secondary. Although it is desirable for the latter two to be small to some extent, too-small values mean insufficient exploration from the original protein. On the other hand, the fact that LBA's performance is high only when the mask ratio is 15% indicates that this model is sensitive to this hyperparameter. Since it is difficult to determine appropriate hyperparameters for the test set in advance, it is not known whether we can realize the best performance of the 15% setting in practice.
> > > > >
> > > > > Thank you for your insightful comments and for acknowledging the importance of LBA as the primary metric, with structural diversity and sequence preservation being secondary considerations. We agree that while minimizing the latter two metrics is desirable to some extent, overly small values could indeed hinder sufficient exploration from the original protein.
> > > > >
> > > > > In response to your concern, we want to emphasize that although our model is not out-of-the-box in terms of sequence diversity, it is not significantly inferior to the baseline models in this regard. Importantly, our model shows superior results in the contact overlap measure of structural preservation and achieves quite competitive results in both the TM-Score and RMSD measures. This demonstrates that while our model excels primarily in LBA, it also performs robustly in maintaining structural integrity and exploring new sequences effectively, thereby providing a well-rounded performance across multiple important metrics.
> > > > >
> > > > > Regarding our model's sensitivity to mask ratio, we recognize the challenge in pre-determining appropriate hyperparameters for the test set. To address this, we conducted ablation studies on mask ratios, ranging from 10\% to 70\%, with a focus on the 15\%-20\% range as part of our hyperparameter tuning on validation sets. These studies help us better understand the impact of different mask ratios, ensuring our model achieves optimal performance across various settings. We provide all trained masked models for users, enabling them to choose sequences with high mask ratios for more exploration or low mask ratios for self-consistent designs in their wet lab experiments.
> > > > >
> > > > > > I want to confirm how the authors compute the similarity between two datasets (e.g., train vs. test.) Does the statement imply that there exists a sequence pair (one from train and one from test) that is similar? If so, in my understanding, this is different from the procedure of existing studies. For example, the following sentence from Yang et al. (2022) implies that there are no similar sequence pairs between the training and test sets: `Importantly, all sequences with >30% identity to the CATH test set were removed from CARP640M's training set in order to obtain a fair evaluation on the CATH test set.`
> > > > >
> > > > > Yes, some pairs of sequences are similar, but assessing sequence similarity alone is insufficient. Firstly, there is a significant disparity in the availability of training data for complexes as compared to sequences alone. For methods like CARP, which train solely on sequences without ligands, there are millions of sequences available (e.g., UniProt, BFD). In contrast, the number of available protein-ligand complexes is much smaller, totaling less than 30,000. Therefore, we must ensure that the number of complexes is sufficient for both training and testing.
> > > > >
> > > > > Secondly, ligands bound to similar sequences are vastly dissimilar (Table 1, Ligand part), and no two similar proteins share similar ligands. Since our study focuses on modeling complexes (including both proteins and ligands), the diversity of ligands outweighs the similarity of protein sequences. This is particularly relevant as a single protein can bind to very distinct ligands.
> > > > >
> > > > > To address these challenges, we employed a time-based splitting approach as described in [1](https://arxiv.org/abs/2202.05146), and filtered out additional similar sequences until the similarity threshold was met. This ensures that our dataset maintains an appropriate balance and diversity for effective modeling of protein-ligand complexes.

---

> ### Author Response · Authors · 2024-07-29
> **Rebuttal by Authors**
>
> > Section 5.2.6 is appropriate for ablation studies of hyperparameters. However, since it selects the mask ratio that gives LBA the best accuracy (i.e., 15%) on the test dataset, I do not think it is an appropriate evaluation of the model (cf. Q.2).
>
> It seems there might be a misunderstanding. In our ablation studies on the impact of masking ratios, we considered mask ratios as part of our hyperparameter tuning on validation sets (Figure 9, blue curve). We selected the mask ratios based on the results from the validation set, not the test set. Coincidentally, the results on the test set and the validation set are quite correlated, as illustrated in Figure 9. Therefore, the chosen mask ratio of 15% was determined based on validation set performance . We no longer choose hyperparameters based on the test set to avoid bias and ensure a more robust evaluation of the model.
>
> > Point (2): Considering the manifold hypothesis and the existence of the adversarial example, the assumption that $ p_{data}(x) \approx p_{data}(x + \delta) $ for any small $\delta$ seems too strong. However, it may be a reasonable assumption if we narrow the assumption and restrict $\delta$ to the one that moves $x$ on the low-dimensional data manifold. We can justify the discussion on Point (2) by changing the domains of the integration from the whole space to the data manifold.
>
> >Point (3): I could not understand the logic of this point (in the following, italics are used for vector values for simplicity.)
> - First, the denominator of the left-hand side of (3), i.e., $\int_{\tilde{x}} q(z_0 \mid \tilde{x})$, is wholly ignored.
> - Second, it is unclear how the following equation is derived:
> $q(z_0 \mid x) = N_x(x \mid z_0 / \mu_0(x), \sigma_0^2(x) / \alpha^2)$.
> - Third, even if the above equation is correct since it is different from the right-hand side of (3), i.e., $N_x(x \mid z_0 / \alpha_0, \sigma_0 / \alpha_0^2)$. I could not understand how the right-hand side in (3) is derived.
>
> We want to stress that the goal of our method is primarily to apply variational diffusion for protein redesign. Deriving new diffusion models is out of the scope of our work. Instead, we effectively utilize the existing diffusion framework proposed by ([2](https://arxiv.org/pdf/2107.00630), [3](https://openreview.net/forum?id=PxTIG12RRHS), [4](https://arxiv.org/abs/2011.13456)). Considering that we add very small perturbations to the coordinates and residue masks in each noising step, the assumption in Point (2) suffices, as it is also done in other established works  ([5](https://arxiv.org/pdf/2304.02198), [6](https://arxiv.org/abs/2301.12485)).
> We acknowledge the mistaken notation in Point 3 which should be $q(x|z_0) = \frac{q(z_0|x)p_{\text{data}}(x)}
> {\int_{\tilde{x}} q(z_0|\tilde{x})p_{\text{data}}(\tilde{x})} \approx
> \frac{q(z_0|x)}
> {\int_{\tilde{x}} q(z_0|\tilde{x})} =
> \mathcal{N}_x (x | z_0 / \alpha_0, \sigma_0^2 / \alpha_0^2 I).$
> We reiterate that our paper aims to implement this established diffusion framework without establishing new diffusion models. The key innovations in protein design include dual denoising domains, preservation of structure with sequence diversity, and a flexible data processing framework. These contributions are significant for the field of biology and protein-drug design as a whole.
>
> - \[1]:(https://arxiv.org/abs/2202.05146)
> - \[2]:(https://arxiv.org/pdf/2107.00630)
> - \[3]:(https://openreview.net/forum?id=PxTIG12RRHS)
> - \[4]:(https://arxiv.org/abs/2011.13456)
> - \[5]:(https://arxiv.org/pdf/2304.02198)
> - \[6]:(https://arxiv.org/abs/2301.12485)

---

### Review · Reviewer_sEMa · 2024-06-05

**Summary Of Contributions:**

This paper focuses on conditional protein design tasks, which simultaneously designs protein sequences and protein structures given ligand SMILES.  To achieve this goal, the proposed model combines techniques from multiple previous publications, such as protein embeddings from ESM-2, generative diffusion procedure from DPL, and feature updating protocol inspired by AlphaFold2. Together with these modules, the proposed model achieves SOTA performance on generating ligand-binding protein with high binding affinities.

**Audience:**

Yes

**Claims And Evidence:**

Yes

**Requested Changes:**

1. Add equation numbers for easier reference.
2. There are some grammar errors in the manuscript such as "this investigation is key to understanding", "vector $z_{sij}$ of ssequences"

**Strengths And Weaknesses:**

- Strengths:
  * This paper propose an algorithm for simultaneously designing protein sequences and 3D structures conditioned on ligand SMILES.

  * The proposed model used a pretrained ESM-2 model to generate protein embeddings and developed a ligand feature embedding method to represent ligand SMILES. The ligand and protein embeddings are converted to both single representation and pair representation for feature updating. Inspired by AlphaFold2, the authors developed feature updating techniques including single representation attention, outer product mean, and triangle multiplicative updates. Using the designed feature updating procedure, the updated embedding is used in a generative diffusion model for conditional protein sequence and structure prediction.

  * The proposed model achieved SOTA performance in generating ligand-binding protein with high affinity.


- Weaknesses:
  * Lack of novelty in methodology.
  * The manuscript lacks some details on the following questions:
    * This paper focuses on a conditional protein generation task given a ligand. Can the authors explain in which practical scenarios that a protein needs to be designed for a given small molecule ligand?

    * How to determine which amino acids to mask?
    * As ProteinReDiff can generate Protein Structure, why is OmegaFold still needed in the evaluation?
    * Is RMSD only computed on backbone atoms?
    * Which baseline models are designed for conditional protein generation given a ligand? Which ones are adapted for this purpose?
    * In the training data, both apo and holo structures are included. For the apo ones, how is the ligand embedding and pair embedding generated?
    * Does DPL results in Table 3 represent the adapted DPL version mentioned in 4.2.4 Efficiency of our innovations?

---

> ### Author Response · Authors · 2024-06-18
> **Rebuttal by Authors**
>
> Thank you so much for your very constructive and useful comments! We appreciate your great input and thank you for helping improve our paper! Below are our point-to-point answers according to your questions and weaknesses.
>
> ## Weaknesses
>
> > Lack of novelty in methodology
>
> While the idea of using diffusion models to estimate the 3D structures of proteins is common in the context of protein design, our study takes this approach further by addressing a new and challenging problem:
>
> 1. **Redesign of high-affinity ligand-binding proteins without detailed structural information**: We show in Table 2 (revised draft) that most protein design models require structure information as input, which can be expensive to characterize from wet lab experiments or unavailable for novel diseases like COVID-19.
>
> 2. **Denoising in both sequence and structural spaces**: Our loss function performs denoising in both sequence and structural spaces, whereas other models typically denoise in only one of these domains. For example, ProteinMPNN, LigandMPNN, and MIF operate in sequence space, while DPL operates in structural space.
>
> 3. **Generation of diverse sequences while preserving structural integrity**: Unlike protein design models such as ProteinMPNN and CARP that prioritize sequence similarity, our model generates diverse sequences while maintaining structural integrity, as measured by contact map overlap.
>
> ## Questions
>
> > This paper focuses on a conditional protein generation task given a ligand. Can the authors explain in which practical scenarios a protein needs to be designed for a given small molecule ligand?
>
> Designing proteins for specific small molecule ligands is crucial in several practical scenarios, including:
>
> - **Drug development**: Creating therapeutic agents (e.g., antibodies) with high specificity and efficacy ([1](https://www.nature.com/articles/s41467-023-38039-x)).
> - **Enzyme engineering**: Optimizing industrial biochemical reactions ([2](https://www.ncbi.nlm.nih.gov/pmc/articles/PMC5474765/)).
> - **Biosensors**: Detecting toxins and metabolites in medical diagnostics and environmental monitoring.
> - **Synthetic biology**: Controlling metabolic pathways and developing targeted drug delivery systems ([3](https://pubmed.ncbi.nlm.nih.gov/35918219/)).
>
> In all of these applications, the ligands are either drugs, metabolites, or small-molecule modulators. This highlights the importance and broad impact of our approach in designing ligand-binding proteins.
>
> > How to determine which amino acids to mask?
>
> As detailed in our paper, we implemented stochastic masking on random segments of the amino acid sequences. The mask segments change with each epoch, enabling the model to learn to reconstruct various parts of the input protein sequence. This approach is akin to the weakly or self-supervised learning method described in [4](https://arxiv.org/pdf/2305.08491) and [5](https://arxiv.org/pdf/2401.00897). We intentionally avoid using prior knowledge to determine the mask regions to prevent biasing our training. For instance, we experimented with masking just pocket regions (6 Angstrom from ligands) for all proteins, but this approach hindered the model's ability to learn the overall structure, resulting in poor reconstruction performance.
>
> > As ProteinReDiff can generate Protein Structure, why is OmegaFold still needed in the evaluation?
>
> While ProteinReDiff is capable of generating both protein sequences and C-alpha backbones, we require a full all-atom protein structure, including side chains, for accurate docking simulations using AutoDock Vina. OmegaFold is utilized to predict the complete all-atom protein structure from the sequences generated by ProteinReDiff. This step ensures that the structures used in docking simulations are as realistic and accurate as possible, thereby providing a more reliable evaluation of ligand-binding affinities.
>
> Additionally, using C-alpha backbones helps better determine docked structures by providing a more precise initial alignment structure. The table below includes the docking scores for ProteinReDiff with 0.05 and 0.15 mask ratios, as well as for the baseline models ESMIF and CARP, the metrics are reported as mean values derived from the dataset's samples:
>
> |Model                  | Aligned (kcal/mol) ↓ | Non-aligned (kcal/mol) ↓ |
> |-----------------------|----------------------|--------------------------|
> | ESMIF                 | -5.555                   | -1.678 |
> | CARP                  | -5.657                   | -1.721 |
> |ProteinReDiff (0.05)   | -5.804                   | -1.062 |
> |ProteinReDiff (0.15)   | -6.803                   | -1.813 |
>
> These results demonstrate that C-alpha alignment improves the accuracy of the docking simulations, thereby enhancing the reliability of our evaluation of ligand-binding affinities.

---

> ### Author Response · Authors · 2024-06-18
> **Rebuttal by Authors**
>
> ## Questions
> > Is RMSD only computed on backbone atoms?
>
> In our evaluation, RMSD is computed on the whole protein structure, not just the backbone atoms. This comprehensive approach ensures that our assessment accurately reflects the structural fidelity of the entire protein, including both the backbone and side chains, thereby providing a more thorough and precise measure of structural deviations.
>
> > Which baseline models are designed for conditional protein generation given a ligand? Which ones are adapted for this purpose?
>
> Among the baseline models, LigandMPNN is the most similar to our task in designing ligand-binding proteins but requires binding pocket information, which is not a requirement for our model. The other baseline models, such as MIF, MIF-ST, ProteinMPNN, The Protein Generator, ESMIF, CARP, and DPL, do not take ligand information and have been adapted for this task to provide a comprehensive performance comparison. We selected these models as they use the most current advanced deep learning models (e.g., transformers, diffusion, etc.) and are high-throughput, capable of generating thousands of sequences with high fidelity. Please refer to Table 2 (revised draft) for a thorough comparison.
>
> Previous ligand-binding protein models are either laboratory-intensive or low-throughput ([6](https://link.springer.com/book/10.1007/978-1-4939-3569-7), [7](https://journals.plos.org/ploscompbiol/article?id=10.1371/journal.pcbi.1008178)). Some works require pocket information which could not be easily obtained for novel targets ([8](https://www.sciencedirect.com/science/article/pii/S0959440X16301464), [9](https://www.biorxiv.org/content/10.1101/2021.01.13.426598v1.full)).
>
> Our model differs significantly from a category of drug-generative models that rely on protein data to produce ligands bound to them. Known as structure-based drug design (SBDD) ([10](https://github.com/zaixizhang/Awesome-SBDD)), these models are heavily explored by research firms and the research community for their pharmaceutical applications. They are unrelated to our objectives and therefore are not included in our comparisons.
>
> > In the training data, both apo and holo-structures are included. For the apo ones, how is the ligand embedding and pair embedding generated?
>
> For apo structures, the ligand SMILES is represented as an asterisk (*), which acts as a dummy placeholder in RDKit's notation for an unspecified atom. This placeholder allows the ligand embedding (atom features) to take on zero or unspecified chemical features such as zero mass, zero charge, and unspecified hybridization. For the pair embedding (bond feature), it is represented as a zero [0] bond. This approach enables the model to focus on learning the internal representation of proteins alone when dealing with apo structures. By using this dummy placeholder, our model can train on both holo and apo structures, enriching the protein representation. This training procedure for complexes is not commonly employed in previous work, making our approach unique and more robust.
>
> > Does the DPL results in Table 3 represent the adapted DPL version mentioned in 4.2.4 Efficiency of our innovations?
>
> Yes, the DPL results presented in Table 3 indeed represent the adapted DPL version discussed in Section 4.2.4, "Efficiency of our innovations". To ensure clarity, we have now explicitly stated this connection in the revised manuscript. You can find the updated information on page 11, Section 4.2.4.
>
> Generally, DPL is not a design model; it's more of a docking model. Thus, we needed to adjust the loss function (with retraining) to suit our task. We believe this clarification will help readers understand the direct link between our described innovations and the results presented.
>
> ### Requested Changes
>
> > Add equation numbers for easier reference.
>
> Done in revised draft.
>
> > There are some grammar errors in the manuscript such as "this investigation is key to understanding", "vector $z_{sij}$ of ssequences".
>
> Done in revised draft.
>
> References:
>
> \[1]: https://www.nature.com/articles/s41467-023-38039-x
>
> \[2]: https://www.ncbi.nlm.nih.gov/pmc/articles/PMC5474765/
>
> \[3]: https://pubmed.ncbi.nlm.nih.gov/35918219/
>
> \[4]: https://arxiv.org/pdf/2305.08491
>
> \[5]: https://arxiv.org/pdf/2401.00897
>
> \[6]: https://link.springer.com/book/10.1007/978-1-4939-3569-7
>
> \[7]: https://journals.plos.org/ploscompbiol/article?id=10.1371/journal.pcbi.1008178
>
> \[8]: https://www.sciencedirect.com/science/article/pii/S0959440X16301464
>
> \[9]: https://www.biorxiv.org/content/10.1101/2021.01.13.426598v1.full
>
> \[10]: https://github.com/zaixizhang/Awesome-SBDD

---

### Review · Reviewer_GfKZ · 2024-06-07

**Summary Of Contributions:**

This paper introduces and evaluates ProteinReDiff, an equivariant diffusion model to generate protein-ligand complexes with high binding affinity (in contrast to most works which just generate proteins or ligands individually). The paper mostly describes the method and shows some experimental results.

**Audience:**

No

**Broader Impact Concerns:**

Besides the exaggeration in the writing, nothing stands out.

**Claims And Evidence:**

No

**Requested Changes:**

TMLR seeks papers which make claims that are 1) interesting 2) well-supported by evidence. As far as I can tell, the claims of this paper are:

1. Propose a new model called "ProteinReDiff"
2. That their framework represents a "significant advancement"

Claim (1) seems obviously true but not interesting (anybody can make a new model). Claim (2) is interesting but seems not well-supported. Therefore, my proposed changes are as follows:

### 1: revise writing

This paper was full of vague statements and embellished language which seem inappropriate for a scientific publication. For example:

- the authors call their contribution a "novel framework", which seems like a clear exaggeration. It is a diffusion model which very closely follows existing diffusion model training strategies and is trained on very similar datasets. A more accurate description might be "novel model"
- A lot of unsupported speculation about model internals. For example, the outer product mean is said to "amalgamate all available data, culminating in coherent representations". What is coherent?? The triangular multiplicative updates are said to "While attention mechanisms help identify residues with significant influence, the utilization of triangular multiplicative updates becomes crucial in preventing excessive focus on specific residue subsequence". What is "excessive focus" and how do you know it is reducing it?
- The authors conclude the paper by calling their method "groundbreaking". AlphaFold2 was groundbreaking. How does your work compare?
-  "Such fluctuations in validation loss, despite its general decline, suggest the model’s adaptive optimization in the face of complex data patterns." This seems very speculative. Is there some sort of theory about how large fluctuations in validation loss provide evidence of learning?

### 2: better discussion of related work

The paper was missing:
- A clear statement about how their diffusion model training scheme compares with previous work. My impression is that it is exactly a standard denoising diffusion model. The authors should comment on differences.
- How does the architecture compare to Nakata et al, whose model seemed to form the basis for ProteinReDiff?
- How do evaluation metrics compare with previous work?

### 3: explain method better

The algorithms describing the method are quite vague. Notations like "LinearNoBias" are less clear than just writing $Wm$ for explicit matrix multiplication. I think a more precise and detailed explanation is required.

### 4: clarity and further analysis of experimental results

The quantitative results in table 3 suggest that the model's performance is very much in line with previous work, except for the one 15% masking entry. This can hardly be described as "groundbreaking". Furthermore, there is a clear upward trend in LBA as the masking increases, with 15% being a notable exception. Especially considering the lack of any error bars, could this just be an outlier or noise? Overall this table constitutes the main evidence to support the paper's claims, so a careful analysis of the results is important.

**Strengths And Weaknesses:**

The abstract of the paper states:

> In this study, we introduce ProteinReDiff, a novel computational framework designed to **revolutionize** the redesign of ligand-binding proteins.

(emphasis my own). Despite proposing a reasonable method, in my opinion the paper fails to reach this lofty goal. The main weaknesses of the paper are:

- Lack of clarity about method in the writing
- Lack of clarity about relationship to related work
- General exaggeration / embellished language everywhere
- Mediocre experimental results

---

> ### Author Response · Authors · 2024-06-27
> **Rebuttal by Authors**
>
> Thank you so much for your very constructive and useful comments! We appreciate your great input and thank you for helping improve our paper! Below are our point-to-point answers according to your requested changes and weaknesses.
>
> ## Weakness
> > General exaggeration / embellished language everywhere
>
> We have revised the manuscript to exclude exaggerated and embellished language, ensuring a more accurate and objective presentation of our work.
>
> ## Requested Changes
>
> ### 1: revise writing
>
> > the authors call their contribution a "novel framework", which seems like a clear exaggeration. It is a diffusion model which very closely follows existing diffusion model training strategies and is trained on very similar datasets. A more accurate description might be "novel model"
>
> While our work builds upon established diffusion model training strategies and utilizes similar datasets, it is specifically tailored for redesigning ligand-binding proteins without heavily relying on detailed structural data. This approach addresses a relatively underexplored area in protein design, which justifies it as a valuable framework. Here are the key points highlighting our contributions:
>
> 1. **Integration of Diverse Datasets**: We have combined two datasets, PDBBind and CATH proteins, in a new way of training. While PDBBind provides ligand information, CATH proteins lack this data. To address this, we assigned the ligand SMILES of CATH proteins as asterisks, representing unknown or dummy molecules. This approach effectively doubles our training data, enabling our model to learn richer protein representations from both structural datasets.
>
> 2. **Redesign without Detailed Structural Information**: Our method enables the redesign of high-affinity ligand-binding proteins without the need for detailed structural information such as pocket and ligand 3D configurations. This is a significant advantage, as many existing protein design models rely on costly wetlab characterization, which may not be feasible for emerging diseases like COVID-19. As demonstrated in **Table 3** of our revised draft, our approach bypasses these requirements, making it more practical for real-world applications.
>
> 3. **Dual-Domain Denoising**: Unlike other models that typically focus on denoising either in sequence or structural spaces alone (e.g., ProteinMPNN, LigandMPNN, MIF in sequence space, DPL in structural space), our approach integrates denoising across both sequence and structural dimensions through a novel loss function. This holistic approach enhances the model's ability to generate accurate and functional protein-ligand complexes.
>
> 4. **Generating Diverse Sequences with Structural Integrity**: Our model excels in generating diverse sequences while preserving structural integrity. In contrast to models like ProteinMPNN and CARP, which prioritize sequence similarity, our framework demonstrates its ability to maintain structural fidelity, as evidenced by strong overlap in contact maps. This balance between diversity and structural preservation is crucial for effective protein design.
>
> In essence, our framework addresses a less-explored yet valuable aspect of protein design by combining innovative data integration, a dual-domain denoising approach, and a focus on practical applicability without the need for extensive structural information. These elements collectively establish our work as a valuable framework in the domain.
>
> > The authors conclude the paper by calling their method "groundbreaking". AlphaFold2 was groundbreaking. How does your work compare?
>
> We acknowledge that describing our method as "groundbreaking" was an exaggeration and embellished language on our part. We revised the manuscript to more accurately reflect the contributions and impact of our work. On the other hand, comparing our model directly to AlphaFold2 is not appropriate because the two models address different tasks. AlphaFold2 is focused on predicting protein structures from sequences, functioning primarily as a supervised learning model. In contrast, our model is designed to generate new protein sequences and complexes, which is a different and complementary task. Additionally, our model operates on a smaller scale than AlphaFold2 due to our limited resources for training and processing data. Despite these differences, we believe that our work on redesigning sequences is valuable and addresses a unique challenge in the field of protein design.
>
> Recognizing the advanced equivariant architecture of AF2, we thus incorporated with adapation 2 AF2 modules which are MSARowAttention and OuterProductMean in **Section 4.2**. We showed that having those models are an enhancement to our model compared to DPL, which does not have those.

---

> ### Author Response · Authors · 2024-06-27
> **Rebuttal by Authors**
>
> ## Requested Changes
>
> ### 1: revise writing
>
> > A lot of unsupported speculation about model internals. For example, the outer product mean is said to "amalgamate all available data, culminating in coherent representations". What is coherent?? The triangular multiplicative updates are said to "While attention mechanisms help identify residues with significant influence, the utilization of triangular multiplicative updates becomes crucial in preventing excessive focus on specific residue subsequence". What is "excessive focus" and how do you know it is reducing it?
>
> To clarify, the Outer Product Mean (OPM) module in our approach draws inspiration from tensor product techniques used in natural language processing (NLP) research ([1](https://aclanthology.org/N18-1114/), [2](https://ojs.aaai.org/index.php/AAAI/article/view/3934)). It leverages the outer product to correlate information between each amino acid pair in a sequence, enabling the model to learn representations that account for pairwise interactions, similar to the approach used in Alphafold 2. This process allows the model to integrate information across the entire protein sequence, resulting in more coherent and comprehensive structural hypotheses. In light of evidence for the effectiveness of model internals, we have now provided related work that explains how this module actually works in various protein design schemes ([3](https://www.frontiersin.org/journals/artificial-intelligence/articles/10.3389/frai.2023.1149748/full), [4](https://www.nature.com/articles/s41392-023-01381-z)).
>
> Regarding the triangular multiplicative update mechanism, it is adapted from Alphafold 2 (AF2) and plays a crucial role in our model's "Folding Block." This module facilitates iterative refinement of structural representations, aiming to avoid overemphasizing specific structural motifs, such as helices or beta sheets, at the expense of flexible loops. The term "excessive focus" refers to the potential bias towards certain structural elements, which the triangular updates mitigate by adhering to geometric constraints (following triangle inequalities).
>
> In our revised manuscript, we have contextualized these concepts, aiming for clearer and more precise explanations. This update can be found in **Section 4.2**.
>
> > "Such fluctuations in validation loss, despite its general decline, suggest the model's adaptive optimization in the face of complex data patterns." This seems very speculative. Is there some sort of theory about how large fluctuations in validation loss provide evidence of learning?
>
> We agree that the assertion regarding fluctuations in validation loss indicating the model's adaptive optimization may seem speculative without additional context. To clarify, while fluctuations in validation loss are not inherently indicative of successful learning, they can occur in the presence of complex data patterns and challenging learning tasks. In our observation, the overall downward trend in validation loss, despite its variability, suggests that the model is gradually improving its generalization capabilities. To address this, we will revise the wording in the manuscript to more accurately reflect the observed behavior. This update has been included in **Section 5.1.3**.
>
> ### 2: better discussion of related work
>
> > A clear statement about how their diffusion model training scheme compares with previous work. My impression is that it is exactly a standard denoising diffusion model. The authors should comment on differences.
>
> We acknowledge that our diffusion model training scheme closely follows the principles of standard denoising/variance-preserving diffusion models [5](https://arxiv.org/pdf/2107.00630). However, we have introduced significant modifications to the loss function to suit the task of ligand-binding protein redesign better. By tailoring the loss function to integrate both sequence and structural spaces, our approach effectively addresses the unique challenges of protein-ligand interactions. This dual-domain denoising strategy distinguishes our work from standard diffusion models and enhances the model's ability to generate high-affinity ligand-binding proteins while maintaining structural fidelity. We have discussed these modifications in detail in **Section 5.1.2** of the revised manuscript.
>
> On the other hand, our contribution does not lie in the training scheme itself. Our primary innovation is the application of diffusion models to the specific task of ligand-binding protein redesign without requiring detailed structural information. This adaptation leverages the strengths of diffusion models to address the unique challenges of protein-ligand interactions.

---

> ### Author Response · Authors · 2024-06-27
> **Rebuttal by Authors**
>
> ### 2: better discussion of related work
> > How does the architecture compare to Nakata et al, whose model seemed to form the basis for ProteinReDiff?
>
> We have updated the manuscript to include details in **Section 4.2** on how our approach deviates from the residual feature update procedure employed in the original DPL model. While the DPL model relied on Alphafold2's Triangular Multiplicative Update for updating single and pair representations, our objective is to optimize this procedure for greater efficiency. Specifically, we incorporate enhancements such as the Outer Product Mean and Single Representation Attention to formulate sequence representational hypotheses for proteins and to model suitable motifs for binding target ligands. These modules, integral to Evoformer, the sequence-based module of AF2, play a crucial role in extracting essential connections among internal motifs that serve structural functions (i.e., ligand binding) when structural information is not explicitly provided during training. These adaptations ensure their effectiveness in capturing the intricate interplay between proteins and ligands.
>
> Additionally, we have made significant modifications to the DPL model to better align it with our objectives. While the DPL model was originally geared towards protein-ligand complex generation, we adapted it for our purposes by modifying loss functions and incorporating a sequence prediction module. These changes enable our model to generate high-affinity ligand-binding proteins without requiring detailed structural information, a key differentiator from the DPL model. This idea has been mentioned in a new paragraph in **Section 5.2.5** on Benchmark Model Selection in the revised manuscript. This section provides a comprehensive comparison of our model with state-of-the-art approaches, particularly those relevant to protein design tasks. It includes a table summarizing the input and output characteristics of various models, highlighting the unique features and capabilities of ProteinReDiff.
>
> > How do evaluation metrics compare with previous work?
>
> Our model excels in generating diverse sequences while preserving structural integrity. Unlike models like ProteinMPNN and CARP, which prioritize sequence similarity, our framework demonstrates a strong ability to maintain structural fidelity, as evidenced by significant overlap in contact maps. While other works have relied on wet lab experiment results, which we do not have access to, we extensively used docking simulations to achieve affinity scores that closely match experimental results. This approach allows us to validate our model's performance effectively despite the lack of direct experimental data.  We have included this detailed comparison in **Section 5.2** of the revised manuscript.
>
> ### 3: explain method better
> > The algorithms describing the method are quite vague. Notations like "LinearNoBias" are less clear than just writing $Wm$ for explicit matrix multiplication. I think a more precise and detailed explanation is required.
>
> We acknowledge the need for clearer and more detailed explanations of the algorithms. LinearNoBias is a notation that originates from AlphaFold2, referring to a linear MLP without a bias term. We adapted this notation to suit our architecture. For example, the MSARowAttention in AlphaFold2 applies only to a single sequence, but we have modified it to work across multiple sequences, introducing the 's' term to indicate multiple sequences. We have provided better explanations of these terms in **Section 4.2** of the revised manuscript. Furthermore, we added significantly related work that explained the AF2 modules better in the protein representation learning context.

---

> ### Author Response · Authors · 2024-06-27
> **Rebuttal by Authors**
>
> ### 4: clarity and further analysis of experimental results
> > The quantitative results in Table 3 suggest that the model's performance is very much in line with previous work, except for the one 15% masking entry. This can hardly be described as "groundbreaking".
>
> We acknowledge that describing our results as "groundbreaking" was an exaggeration. We have revised the manuscript to use more accurate language to describe our contributions.
>
> > Furthermore, there is a clear upward trend in LBA as the masking increases, with 15% being a notable exception. Especially considering the lack of any error bars, could this just be an outlier or noise?
>
> We acknowledge that the 15% masking entry stands out as an exception in the upward trend observed in Ligand Binding Affinity (LBA) with increasing masking levels. This exception can be attributed to the unique balance it achieves between beneficial modifications and maintaining the functional precision of the protein. While other masking ratios within ProteinReDiff exhibit a range of effectiveness, lower masking ratios, although still performing better than baseline models, do not reach the peak performance achieved at the 15% masking level. Conversely, higher masking ratios tend to introduce too many modifications, which disrupt the protein's functional precision and diminish the model's ability to enhance binding affinity effectively. This highlights the importance of optimizing the masking ratio: the 15% level appears to strike the optimal balance, introducing beneficial modifications while maintaining the necessary functional integrity of the protein. We have included this idea in **Section 5.2.5** of the revised manuscript.
>
> Additionally, we have now included margins of error in **Table 4** to provide better clarity on the variability of the results and to support a more robust analysis of our findings. We see that our 15% masked model are on par with other top-scoring models in structural fidelity and affinity score.
>
> > Overall this table constitutes the main evidence to support the paper's claims, so a careful analysis of the results is important.
>
> We agree that a careful analysis of the results is crucial. We will provide a thorough evaluation and statistical analysis in **Section 5.2.5** to strengthen our claims in the revised manuscript.
>
> References:
>
> \[1]: https://aclanthology.org/N18-1114/
>
> \[2]: https://ojs.aaai.org/index.php/AAAI/article/view/3934
>
> \[3]: https://www.frontiersin.org/journals/artificial-intelligence/articles/10.3389/frai.2023.1149748/full
>
> \[4]: https://www.nature.com/articles/s41392-023-01381-z
>
> \[5]: https://arxiv.org/pdf/2107.00630

---

> > ### Comment · Reviewer_GfKZ · 2024-07-02
> > **Revisions are helpful, a few further questions**
> >
> > Thank you for the revisions. I think the manuscript has improved significantly (although I have not had the time to examine it in great detail).
> >
> > My main unresolved question is the related work. Currently I am having trouble understanding the key differences. You've mentioned a lot: the loss, the dataset, a focus on diversity, but this is not summarized very clearly. Furthermore, a lot of your new related work discussion is in section 5 (experiments), which does not seem like an appropriate place for it.
> >
> > Would you consider adding a clear related work section which answers "how does the model/experiments in this work build upon and differ from previous work"?

---

> > > ### Author Response · Authors · 2024-07-09
> > > **Related works added**
> > >
> > > Thanks for your feedback! To clarify the key differences of our model, we just added Section 2.3 for relevant works on protein redesign. We also highlighted what could be improved from previous models.

---

### Comment · Action_Editor_ozrY · 2024-07-07
**Extended reply period**

Dear reviewers,

The author contacted me asking for extension of the reply period due to the computational time they need for their reply.
We agreed in  7/7/2024 as the new deadline. Which is today. Please allow the authors a few more days!

Dear authors,

Please post your reply as soon as possible!

---

> ### Author Response · Authors · 2024-07-08
> **Extended reply period**
>
> Dear Action Editor and Reviewers,
>
> We appreciate your extension to 7/7/2024. We just got finished with ablation studies. We found several more interesting results and would like to integrate them into our work. Please allow within 24 hours from now to polish our work.
>
> Thank you for your understanding as we're wrapping up!

---

### Decision · Action_Editor_ozrY · 2024-08-02

**Recommendation:** Reject

**Comment:**

Neither Reviewer dYXK nor GfKZ assured that the performance peak at the 15% masking is reproducible, dYXK also questions the hyperparameter selection procedure resulting in using the 15% masking result in their question 6.
Additionally the very strong difference between test and train performance in the docking energies on Figure 9 require further investigation.

Docking based score is known to be very noisy, and require expertise in preparing the protein to get energies correlated to real affinities in any meaningful way. Also generally accepted practice to only consider possible binding candidates under -6kcal/mol, meaning that most of the energies presented in the paper correspond to too week binding to be trustable.

**Audience:**

Generative protein models are interesting for the audience of the present journal. The focus of this paper is still quite strongly on the computational biology side, reformulating it more towards the ML audience would be beneficial.

**Claims And Evidence:**

Not all reviewers agree that convincing and clear evidence was provided. The only metric outperformed is a Docking based proxy of the affinity. It is well known to be very noisy. Multiple reviewers question the sudden increase at the 15% masking rate, their opinion is that methodological error cannot be ruled out.

**Resubmission Of Major Revision:**

The authors may consider submitting a major revision at a later time.